psychology/cognition/neuroscience

emotion recognition, vocal emotions, speech prosody, socio-emotional adjustment, children

**Author for correspondence:**
César F. Lima
e-mail: cesar.lima@iscte-iul.pt

†Joint first authors.

# Associations between vocal emotion recognition and socio-emotional adjustment in children

Leonor Neves[1,†], Marta Martins[1,†], Ana Isabel Correia[1], São Luís Castro[2] and César F. Lima[1,3]

[1]Centro de Investigação e Intervenção Social (CIS-IUL), Instituto Universitário de Lisboa (ISCTE-IUL), Av. das Forças Armadas, 1649-026 Lisboa, Portugal
[2]Centro de Psicologia da Universidade do Porto (CPUP), Faculdade de Psicologia e de Ciências da Educação da Universidade do Porto (FPCEUP), Porto, Portugal
[3]Institute of Cognitive Neuroscience, University College London, London, UK

CFL, 0000-0003-3058-7204

The human voice is a primary channel for emotional communication. It is often presumed that being able to recognize vocal emotions is important for everyday socio-emotional functioning, but evidence for this assumption remains scarce. Here, we examined relationships between vocal emotion recognition and socio-emotional adjustment in children. The sample included 141 6- to 8-year-old children, and the emotion tasks required them to categorize five emotions (anger, disgust, fear, happiness, sadness, plus neutrality), as conveyed by two types of vocal emotional cues: speech prosody and non-verbal vocalizations such as laughter. Socio-emotional adjustment was evaluated by the children's teachers using a multidimensional questionnaire of self-regulation and social behaviour. Based on frequentist and Bayesian analyses, we found that, for speech prosody, higher emotion recognition related to better general socio-emotional adjustment. This association remained significant even when the children's cognitive ability, age, sex and parental education were held constant. Follow-up analyses indicated that higher emotional prosody recognition was more robustly related to the socio-emotional dimensions of prosocial behaviour and cognitive and behavioural self-regulation. For emotion recognition in non-verbal vocalizations, no associations with socio-emotional adjustment were found. A similar null result was obtained for an additional task focused on facial emotion recognition. Overall, these results support the close link between children's emotional prosody recognition skills and their everyday social behaviour.

# 1. Introduction

We perceive emotional information through multiple communication channels, including vocal and facial expressions. These channels offer a window into the emotions of others, and the ability to recognize the conveyed states is an integral part of everyday communication. Although most research has focused on facial expressions, the human voice is a major source of emotional information that reflects a primitive and universal form of communication [1,2]. We can communicate vocal emotions via linguistic information but also via non-verbal cues. Hearing a scream, for instance, might indicate that someone needs help or that there is a threat nearby. Non-verbal emotional cues in the human voice can be divided into two domains: inflections in speech, so-called emotional prosody; and purely non-verbal vocalizations, such as laughter and crying, often called affective bursts (e.g. [3]).

Emotional prosody corresponds to suprasegmental and segmental modifications in the spoken language during emotion episodes. Prosodic cues include pitch, loudness, *tempo*, rhythm and timbre, as embedded in linguistic content [4,5]. Purely non-verbal vocalizations, on the other hand, do not contain any linguistic information (e.g. screams, laughter), and they represent a more primitive form of communication, sometimes described as the auditory equivalent of facial expressions [6]. Prosody and non-verbal vocalizations rely on partly distinct articulatory and perceptual mechanisms [7,8]. Based primarily on studies with adults, we know that listeners can accurately identify several positive and negative emotions from the two types of vocal emotional cues, even when they are heard in isolation and without contextual information (e.g. [9–12]). But it has also been shown that emotion recognition accuracy is higher for non-verbal vocalizations compared with prosody [13–15].

In development, soon after birth, infants can discriminate emotional expressions in non-verbal vocalizations (e.g. [16]) and prosodic cues (e.g. [17]). Emotion recognition abilities improve throughout childhood and adolescence, although it is still not established when they peak [15,18–20]. Infants and young children also show a general preference for auditory over visual information (e.g. tones versus lights, [21]; natural sounds versus pictures, [22]), which might extend to emotional cues. For instance, Ross *et al.* [23] observed that children under the age of eight find it challenging to ignore vocal emotional cues in multimodal stimuli, even if explicitly asked to base their judgement on body cues alone.

Even though it is typically presumed that vocal emotion recognition skills are crucial for communication at any age, research has primarily focused on more basic acoustic, perceptual and neurocognitive aspects of these expressions (e.g. [3,5]). Evidence for associations with broader aspects of everyday socio-emotional functioning remains relatively scarce, particularly in normative samples. Socio-emotional functioning has been defined as a multidimensional and broad concept [24]. It includes the ability to understand our own and others' emotions, to regulate our own behaviour and to establish and maintain relationships [25,26]. These processes start to develop early in life and are linked to health outcomes and well-being [27,28].

Studies on clinical populations are suggestive of a link between vocal emotional processing and socio-emotional functioning, both in adult (e.g. [29–31]) and paediatric samples [32–34]. For instance, youth with severe mood dysregulation and bipolar disorder [32], and with depressive symptoms [33], show impaired recognition of emotional prosody. There are fewer studies on healthy samples, but they point in the same direction. Carton *et al.* [35] showed that better emotional prosody recognition was associated with better self-reported relationship well-being in healthy adults, even after controlling for depressive symptoms. Terracciano *et al.* [36] also found that better emotional prosody recognition correlated with self-reported openness to experience, a trait linked to social behaviour engagement (e.g. [37,38]). We have shown that the ability to recognize laughter authenticity is associated with higher empathic concern and trait emotional contagion in adults [39]. However, there are also null results regarding vocal emotion recognition and traits associated with social behaviour, such as agreeableness and extraversion [40].

Children, like adults, make use of vocal emotions in social interactions, and it is important to understand how this relates to their socio-emotional adjustment, given that childhood is a pivotal period for socio-emotional development [24,25]. Studies with pre-schoolers found that higher emotional prosody recognition correlates with higher peer-rated popularity and lower teacher-rated emotional/behavioural problems [41], as well as with lower parent-rated hyperactivity and conduct problems [42]. Studies with school-age children have also documented associations between emotional prosody recognition and socio-emotional variables including self-reported social avoidance and distress [43], teacher-rated social competence [44,45] and emotional and behavioural difficulties [46],

and peer-rated popularity ([44]; see also [47]). However, some of the identified associations are limited to particular groups (e.g. observed for girls, but not for boys; [41,44]), and null results have been reported too. For instance, pre-schoolers' emotional prosody recognition did not correlate with teacher-rated externalizing problems [41] and parent-rated internalizing behaviour [42]. Additionally, inferences have often been based on relatively small samples, typically less than 80 children, and the focus has been on prosody, leaving the other domain of vocal emotional cues—purely non-verbal vocalizations—unexplored. To our knowledge, only one study included non-verbal vocalizations, and the emphasis was on how children matched vocal with facial information [48]. Other poorly understood questions are whether associations between vocal emotion recognition and socio-emotional functioning are specific and direct, or a consequence of general differences in cognitive abilities and socio-economic background. These general factors correlate with emotion recognition abilities (e.g. [49,50]) and social functioning (e.g. [51–53]), and they are often not considered as potential confounds (e.g. [42,44]).

In the current study, we asked whether vocal emotion recognition relates to socio-emotional adjustment in 6- to 8-year-old children. We covered emotional speech prosody and non-verbal vocalizations, and hypothesized that higher emotion recognition accuracy would be associated with better socio-emotional functioning. If children with a greater ability to recognize emotions from vocal cues are better at interpreting social information, this could favour everyday socio-emotional functioning outcomes, such as the willingness to be friendly and helpful with others, and the ability to stay calm and focused. Participants completed forced-choice emotion recognition tasks focused on the two types of vocal emotional cues. Their teachers were asked to evaluate children's socio-emotional functioning using The Child Self-Regulation and Behaviour Questionnaire (CSBQ; [54]). This is a multidimensional measure, which allows for an analysis of several socio-emotional dimensions (e.g. sociability, prosocial behaviour and emotional self-regulation), and it correlates with outcomes such as peer relationship problems and emotional symptoms [54]. We predicted that children scoring higher on vocal emotion recognition would be rated by their teachers as more socio-emotionally competent in general. We also examined whether this putative association was limited to a particular group of participants (e.g. girls), or driven by general cognitive and socio-economic factors. In other words, we tested if results remained significant when individual differences in age, sex, cognitive ability and parental education are accounted for. This is relevant, considering the reviewed evidence that results can be distinct as a function of sex and age, and that cognitive and socio-economic factors can be associated with emotion recognition and social functioning, therefore being potential confounds.

More exploratory questions asked which socio-emotional functioning dimensions are more clearly linked to vocal emotion recognition, and whether associations between emotion recognition and social-emotional functioning are specific to the auditory domain, or are similarly seen across sensory modalities. In addition to the two vocal emotion recognition tasks, children also completed an emotion recognition task that focused on facial expressions. There is some evidence that better facial emotion recognition relates to fewer behavioural problems [41,42,46] and better self-regulation skills in children [55,56]. But null results have also been reported, namely regarding social avoidance and distress [43] and peer popularity [44]. Moreover, studies that include the two sensory modalities (i.e. vocal and facial emotions) are relatively rare, and they have also reported mixed findings (e.g. [43]).

# 2. Method

## 2.1. Participants

One hundred and forty-eight children were recruited from elementary public schools in a metropolitan area in northern Portugal (Porto). Seven were excluded due to neurological diseases ($n = 2$), atypically low general cognitive ability (Ravens' score < 25th percentile; $n = 4$), or lack of data regarding the socio-emotional measure ($n = 1$). The final sample included 141 children (73 boys) between 6 and 8 years of age ($M = 7.14$ years, s.d. = 0.51, range = 6.34–8.89). They were second graders from seven different classes, each with one teacher assigned for the entire year. All children were Portuguese native speakers and, according to parent reports, had normal hearing and no neurological/ neurodevelopmental disorders (e.g. autism spectrum disorders). Parents' education varied from four to 19 years ($M = 10.98$; s.d. = 3.46). Participants were tested as part of a longitudinal project looking at the effects of music training on emotion recognition and socio-emotional behaviour.

An *a priori* power analysis with G*Power 3.1 [57] indicated that a sample size of at least 138 would be required to detect correlations of $r = 0.30$ or larger between variables, considering an alpha level of 0.05 and a power of 0.95. For regression models including five predictors (age, sex, parental education, general cognitive ability and emotion recognition), a sample of at least 134 participants would be required to detect partial associations of $r = 0.30$ or larger between each predictor variable and socio-emotional adjustment.

This study was approved by the local ethics committee, Iscte—University Institute of Lisbon (reference 28/2019), and it was conducted in agreement with the Declaration of Helsinki. Written informed consent was obtained for all participants from a parent or legal guardian, and children gave verbal assent to participate.

## 2.2. Materials

### 2.2.1. Emotion recognition tasks

The children completed three emotion recognition tasks. Two of them were focused on vocal emotions, speech prosody and non-verbal vocalizations, and the third one on facial expressions. Each task included 60 trials, with 10 different stimuli for each of the following categories: anger, disgust, fear, happiness, sadness and neutrality. The stimuli were part of validated corpora (speech prosody, [9]; non-verbal vocalizations, [11]; facial expressions, Karolinska Directed Emotional Faces database, [58]) that have been frequently used (e.g. [31,59–64]). Speech prosody stimuli were short sentences ($M = 1473$ ms, s.d. = 255) with emotionally neutral semantic content (e.g. 'O quadro está na parede', *The painting is on the wall*), produced by two female speakers to communicate emotions with prosodic cues alone. Non-verbal vocalizations consisted of brief vocal sounds ($M = 966$ ms, s.d. = 259) without linguistic content, such as laughs, screams or sobs, and were produced by two adult female and two adult male speakers. Facial expressions consisted of colour photographs of male and female actors without beards, moustaches, earrings, eyeglasses or visible make-up. Each photograph remained visible until participants responded. Based on validation data from adults, the average recognition accuracy for the stimuli used here was expected to be high (emotional prosody: 78.42%; non-verbal vocalizations: 82.20%; facial expressions: 82.98%).

Participants made a six-alternative forced-choice decision for each stimulus in each of the three tasks. They were asked to identify the expressed emotion from a list that included *neutrality, anger, disgust, fear, happiness* and *sadness*. To improve children's engagement throughout the task, an emoji illustrating each emotional category was included on the response pad and on the laptop screen (visible after the stimulus's offset). Visual aids like emojis or pictures are typically used in vocal emotion recognition tasks intended for children (e.g. [15,18,60]). Each task started with six practice trials (one per emotional category), during which feedback was given. After these trials, the stimuli were presented randomly across two blocks of 30 trials each (no feedback was given). Short pauses were allowed between blocks to ensure that children remained focused and motivated. Each task took approximately 12 min. The tasks were implemented using SuperLab Version 5 (Cedrus Corporation, San Pedro, CA), running on an Apple MacBook Pro laptop. Responses were collected using a seven-button response pad (Cedrus RB-740). Auditory stimuli were presented via headphones (Sennheiser HD 201).

The percentage of correct answers was calculated for each emotional category and task. Accuracy rates were then corrected for response biases using unbiased hit rates, or *Hu*, which were used for all analyses ([65]; for a discussion of biases in forced-choice tasks see, e.g. [66]). Hu values represent the joint probability that a given emotion will be correctly recognized (given that it is presented), and that a given response category will be correctly used (given that it is used at all), such that they vary between 0 and 1. Hu = 0 when no stimulus from a given emotion is correctly recognized, and Hu = 1 when all the stimuli from a given emotion are correctly recognized (e.g. sad prosody), and the corresponding response category (sadness) is always correctly used (i.e. when there are no false alarms). Primary analyses were conducted using average scores for each task because we had no predictions regarding specific emotions.

### 2.2.2. Socio-emotional adjustment

The Child Self-Regulation and Behaviour Questionnaire (CSBQ) is a 33-item educator report (or parent report) questionnaire that assesses children's socio-emotional behaviour [54]. Scale items cover seven

subscales: sociability (seven items, e.g. *Chosen as a friend by others*), externalizing problems (five items, e.g. *Aggressive to children*), internalizing problems (five items, e.g. *Most days distressed or anxious*), prosocial behaviour (five items, e.g. *Plays easily with other children*), behavioural self-regulation (six items, e.g. *Waits their turn in activities*), cognitive self-regulation (five items, e.g. *Persists with difficult tasks*) and emotional self-regulation (six items, e.g. *Is calm and easy-going*). Items are rated on a scale from 1 (*not true*) to 5 (*certainly true*). Individual item scores are then summed to produce total scores for each subscale [54]. A global socio-emotional functioning score was also computed by averaging the means of the seven subscales, hereafter referred to as the *general socio-emotional index*. For this purpose, scores for the externalizing and internalizing problems subscales were reversed so that higher scores indicated better socio-emotional adjustment across all subscales.

The CSBQ translation to European Portuguese followed the guidelines for adapting tests into multiple languages (e.g. [67]). Two European Portuguese native speakers independently translated the items of the original English CSBQ. They were fluent in English, and one of them (C.F.L.) is experienced in the adaptation of questionnaires and an expert in emotion processing. A single version of the questionnaire was obtained by sorting out the disagreements between the two translators. This version was then shown to two laboratory colleagues for a final check on language clarity and naturalness and to discuss the matching between the original and the translated version.

The original CSBQ has sound psychometric properties [54], and in the current dataset, internal consistency values were good to excellent (Cronbach's $\alpha = 0.85$ for general socio-emotional index, ranging from $\alpha = 0.80$ for externalizing/internalizing problems to $\alpha = 0.91$ for cognitive self-regulation).

### 2.2.3. General cognitive ability

The Raven's Coloured Progressive Matrices were used as a measure of general non-verbal cognitive ability [68]. All participants of the final sample performed within the normative range (greater than or equal to 14 out of 36, $M = 22.63$, s.d. $= 4.53$, range $= 14$–33; norms for Portuguese second graders; [69]).

## 2.3. Procedure

Children were tested individually in a quiet room at their school, in two experimental sessions lasting about 45 min in total. General cognitive ability was assessed in the first session and emotion recognition in the second one. The order of the emotion recognition tasks was counterbalanced across participants. Before the sessions, a parent completed a background questionnaire that asked for information about parental education and employment, and the child's history of health issues, such as psychiatric, neurological/neurodevelopmental disorders and hearing impairments.

The CSBQ questionnaire was completed by the children's teacher. Having the teacher completing the questionnaire, instead of a parent, allowed us to maximize sample size, as it could be difficult to get all the 141 parents to return the questionnaire in a timely manner and to minimize social desirability (for a similar approach, e.g. [44,46]). Additionally, many of the CSBQ items focus on interactions with peers and behaviours in the school context, which can be best documented by teachers. The teachers were blind to the hypothesis of the study. They had known the children for about one and a half years when they filled the questionnaire, having had the opportunity to interact with them and observe their behaviour on a daily basis.

## 2.4. Data analysis

The data were analysed using standard frequentist *and* Bayesian analyses conducted with JASP v. 0.14.1 [70]. A repeated-measures analysis of variance (ANOVA) with the task (speech prosody, non-verbal vocalizations and facial expressions) as a within-subjects factor was performed to examine differences in emotion recognition across tasks. Pearson correlations and multiple regression analyses were used to test for associations between our variables of interest. Holm–Bonferroni corrections for multiple comparisons were applied to $p$-values, except in the case of follow-up exploratory analyses (focused on specific emotions and specific dimensions of socio-emotional adjustment), for which uncorrected $p$-values are reported. In addition to $p$-values, a Bayes factor ($BF_{10}$) statistic was estimated for each analysis using the default priors (correlations, stretched beta prior width $= 1$; $t$-tests, zero-centred Cauchy prior with scale parameter 0.707; linear regressions, Jeffreys–Zellner–Siow (JZS) prior of $r = 0.354$; repeated-measures ANOVAs, zero-centred Cauchy prior with a fixed-effects scale factor of $r = 0.5$, a random-effects scale factor of $r = 1$ and a covariates scale factor of $r = 0.354$). Bayes factors

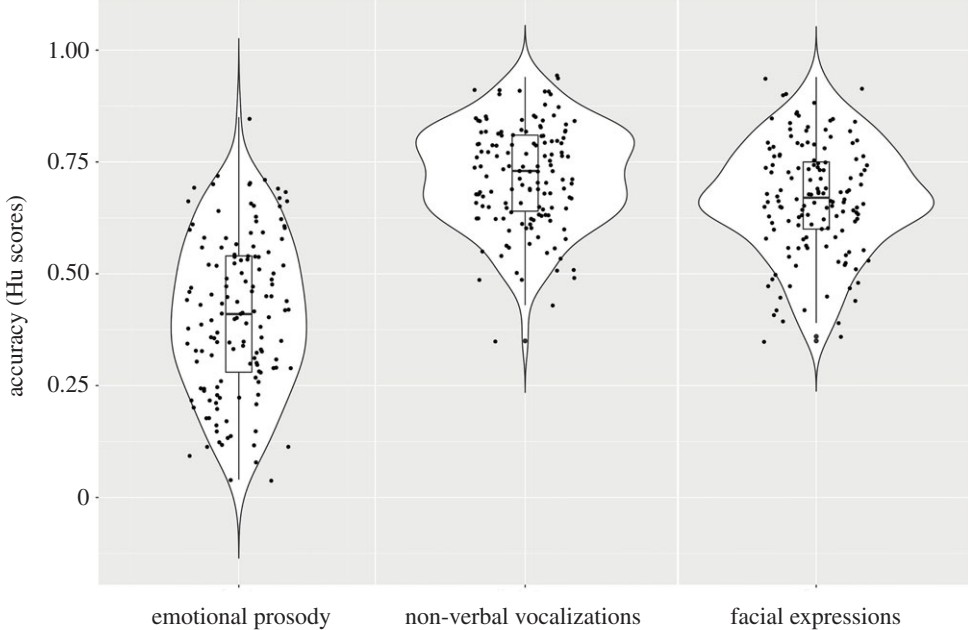

**Figure 1.** Individual results, box plots and violin plots depicting average emotion recognition scores (Hu) for emotional prosody, non-verbal vocalizations and facial expressions.

consider the likelihood of the observed data given the alternative and null hypotheses. $BF_{10}$ values were interpreted according to Jeffreys' guidelines [71,72], such that values below 1 correspond to evidence in favour of the null hypothesis: values between 0.33 and 1 correspond to anecdotal evidence, between 0.10 and 0.33 to substantial evidence, between 0.03 and 0.10 to strong evidence, between 0.01 and 0.03 to very strong evidence, and less than 0.01 to decisive evidence. Values above 1 correspond to evidence for the alternative hypothesis: values between 1 and 3 correspond to anecdotal evidence, between 3 and 10 to substantial evidence, between 10 and 30 to strong evidence, between 30 and 100 to very strong evidence, and greater than 100 to decisive evidence. An advantage of Bayesian statistics is that they allow us to interpret null results and to draw inferences based on them.

The full dataset can be found here: https://osf.io/qfp83/?view_only=47031990843a4897 8ca8058e98118805.

# 3. Results

## 3.1. Emotion recognition

Figure 1 shows children's accuracy in the emotion recognition tasks (see electronic supplementary material, table S1 for statistics for each emotion and electronic supplementary material, table S2 for confusion matrices). Average Hu scores were 0.41 for speech prosody (s.d. = 0.18; range = 0.04–0.85), 0.72 for vocalizations (s.d. = 0.11; range = 0.35–0.94) and 0.67 for faces (s.d. = 0.13; range 0.35–0.94). Performance was above the chance level (0.17) for all three modalities, $p$s < 0.001, $BF_{10}$ > 100, and there was no substantial departure from normality (skewness, range = −1.38–0.75; kurtosis, range = −1.36– 2.64; [73]). A repeated-measures ANOVA with task as within-subjects factor showed that performance differed significantly across tasks, $F_{2,280}$ = 296.48, $p$ < 0.001, $\eta^2$ = 0.68; $BF_{10}$ > 100. It was lowest for prosody (prosody versus vocalizations, $p$ < 0.001, $BF_{10}$ > 100; prosody versus faces, $p$ < 0.001, $BF_{10}$ > 100) and highest for vocalizations (vocalizations versus faces, $p$ < 0.001, $BF_{10}$ > 100). There was a positive correlation between the two vocal emotion recognition tasks ($r$ = 0.32, $p$ < 0.001, $BF_{10}$ > 100), and between these and the faces task (prosody and faces, $r$ = 0.40, $p$ < 0.001, $BF_{10}$ > 100; vocalizations and faces, $r$ = 0.32, $p$ < 0.001, $BF_{10}$ > 100).

## 3.2. Socio-emotional adjustment

Scores for the general socio-emotional index and for each CSBQ subscale are presented in figure 2. The general socio-emotional score was 3.75 on average, and it varied widely among children, from 2.27 to

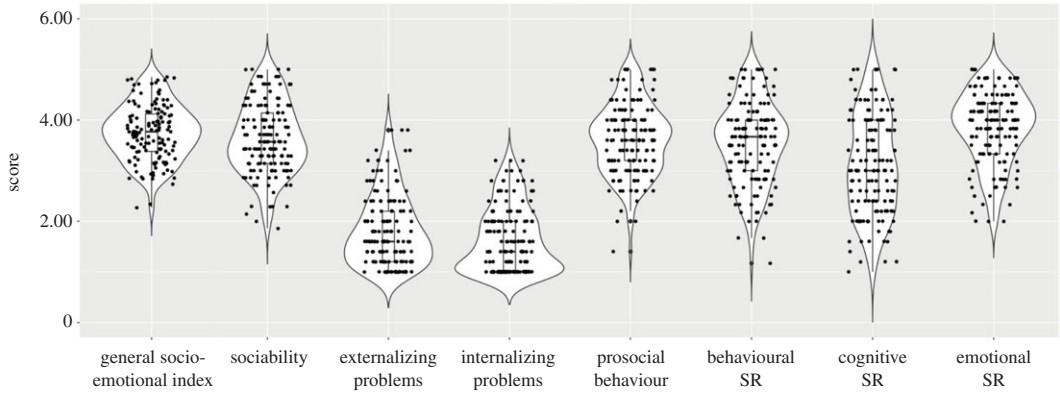

**Figure 2.** Individual results, box plots and violin plots depicting teacher reports on children's social-emotional adjustment, as assessed with the CSBQ questionnaire. SR = self-regulation.

**Table 1.** Associations between the main study variables (emotion recognition and general socio-emotional adjustment) and age, sex, parental education and general cognitive ability. $N = 141$ for all analyses, except for those involving parental education, where $n = 139$. $BF_{10}$ values are indicated in italics. For age, parental education and cognitive ability, values represent Pearson correlation coefficients; for sex, they represent $t$-values (two-tailed independent sample $t$-tests). $^*p < 0.05$; $^{**}p < 0.01$; $^{***}p < 0.001$ (Holm–Bonferroni-corrected).

|  | age | sex | parental education (years) | cognitive ability |
|---|---|---|---|---|
| emotion recognition |  |  |  |  |
| emotional prosody | 0.00 | 0.21 | 0.25* | 0.27* |
|  | *0.11* | *0.18* | *8.05* | *22.14* |
| non-verbal vocalizations | 0.14 | −0.63 | 0.10 | 0.02 |
|  | *0.43* | *0.22* | *0.21* | *0.11* |
| facial expressions | 0.05 | −1.97 | 0.10 | 0.10 |
|  | *0.13* | *1.06* | *0.22* | *0.21* |
| general socio-emotional index | −0.32** | −2.97* | 0.42*** | 0.22 |
|  | *>100* | *9.45* | *>100* | *3.44* |

4.85 (s.d. = 0.55). There was no substantial departure from normality in the CSBQ data (skewness, range = −0.63–0.86; kurtosis, range = −0.84–0.05; electronic supplementary material, table S3; [73]). There were correlations among the CSBQ subscales (see electronic supplementary material, tables S4 and S5), as expected according to the published data [54].

## 3.3. Cognitive and socio-demographic variables

Table 1 shows correlations between the main study variables—emotion recognition and general socio-emotional adjustment—and age, sex, parental education and cognitive ability. Emotion recognition was not associated with demographic or cognitive variables, except for small correlations between emotional prosody recognition and parental education and cognitive ability. Socio-emotional adjustment was higher for girls compared with boys, and it was also higher for younger children and for those with higher parental education.

## 3.4. Emotion recognition and socio-emotional adjustment

In line with our prediction, we found decisive evidence for a correlation between higher emotion recognition in speech prosody and better general socio-emotional adjustment, $r = 0.32$, $p < 0.001$, $BF_{10} > 100$. A similar correlation was not found for emotion recognition in non-verbal vocalizations, however, $r = 0.10$, $p = 0.24$. It was also not found for faces, $r = 0.12$, $p = 0.33$. For both vocalizations and

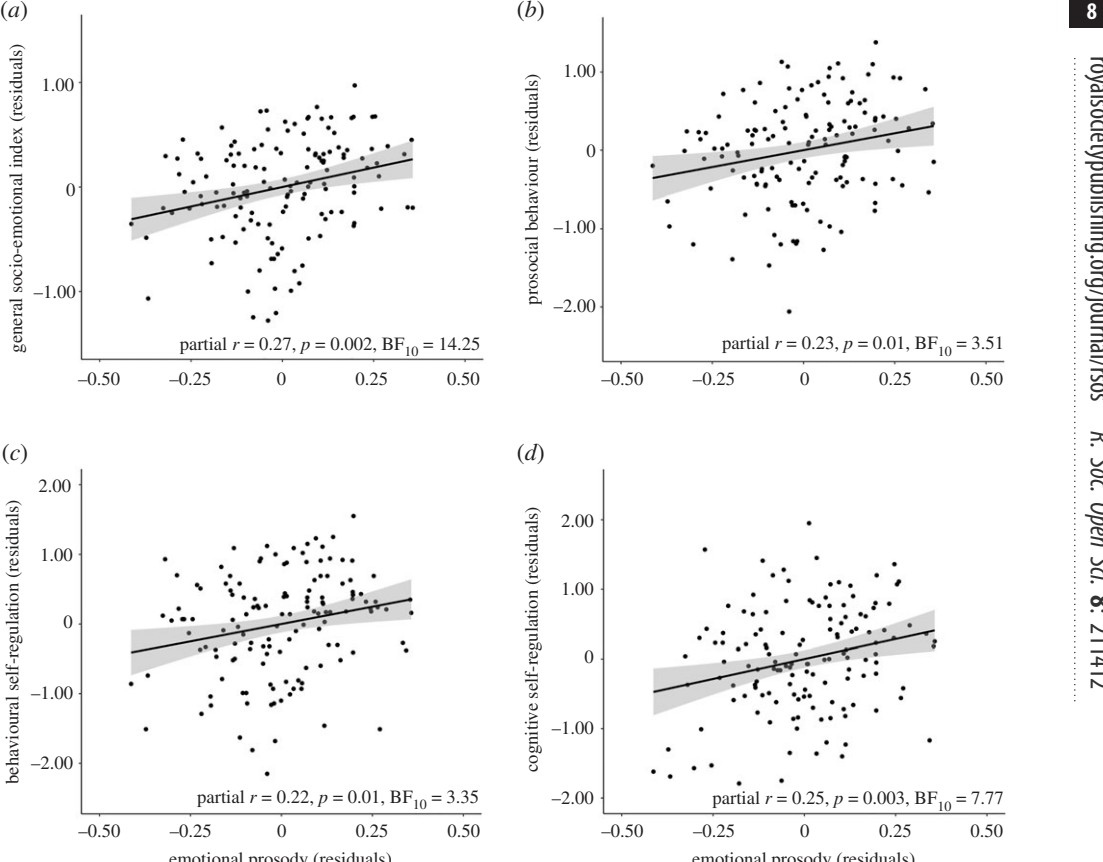

**Figure 3.** Partial regression plots illustrating the relationship between emotion recognition in speech prosody and general socio-emotional adjustment scores (*a*), prosocial behaviour (*b*), behavioural self-regulation (*c*) and cognitive self-regulation (*d*), after removing the effects of age, sex, parental education and cognitive ability. Grey shades represent 95% confidence intervals.

faces, Bayesian analyses provided substantial evidence for the null hypothesis (vocalizations, $BF_{10} = 0.21$; faces, $BF_{10} = 0.27$).[1]

To exclude the possibility that the association between emotional prosody recognition and socio-emotional adjustment was due to cognitive or socio-demographic factors, we used multiple regression. We modelled socio-emotional adjustment scores as a function of age, sex, parental education, cognitive ability and average accuracy on the emotional prosody recognition task. This model explained 30.77% of the variance, $R = 0.58$, $F_{5,133} = 13.26$, $p < 0.001$, $BF_{10} > 100$. Independent contributions were evident for age, partial $r = -0.30$, $p < 0.001$, $BF_{10} = 49.10$; sex, partial $r = 0.22$, $p = 0.01$, $BF_{10} = 3.06$; and parental education, partial $r = 0.28$, $p = 0.001$, $BF_{10} = 28.68$, but not for cognitive ability, $p = 0.34$, $BF_{10} = 0.17$. Crucially, emotional prosody recognition made an independent contribution to the model, partial $r = 0.27$, $p = 0.002$, and the Bayesian analysis provided strong evidence for this contribution, $BF_{10} = 14.25$. We calculated Cook's values and confirmed that this effect was not explained by extreme data points on the regression model (Cook's distance $M = 0.01$, s.d. = 0.01, range = 0.00–0.07). The partial association between emotional prosody recognition and socio-emotional adjustment is illustrated in figure 3*a*.

Although we had no predictions regarding specific emotions, we wanted to ensure that the association between prosody recognition and socio-emotional adjustment was not driven by a single or small subset of emotions. Follow-up multiple regression analyses, conducted separately for each emotion, showed that positive partial correlations could be seen for most emotions, at significant or trend level: happiness, $r = 0.23$, $p = 0.01$, $BF_{10} = 3.81$; anger, $r = 0.22$, $p = 0.01$, $BF_{10} = 3.20$; fear, $r = 0.21$, $p = 0.01$, $BF_{10} = 2.26$; and neutrality, $r = 0.19$, $p = 0.03$, $BF_{10} = 1.24$. For sadness and disgust, the trend

---

[1]Because there was no substantial departure from normality in the data, our analyses were based on untransformed Hu values. However, the pattern of results remained similar when the models were repeated on arcsine-transformed values [65], as can be seen in the electronic supplementary material, Analyses.

was in the same direction but did not reach significance: sadness, $r = 0.12$, $p = 0.16$, $BF_{10} = 0.30$; disgust, $r = 0.13$, $p = 0.12$, $BF_{10} = 0.36$. For completeness, an additional multiple regression was conducted including all emotions simultaneously (see electronic supplementary material, table S6), and none of them contributed uniquely to socio-emotional outcomes ($p_s > 0.34$), probably because of the shared variance across them.

## 3.5. Socio-emotional adjustment dimensions

We also explored how emotional prosody recognition related to specific socio-emotional dimensions, considering the CSBQ subscales: sociability, externalizing problems, internalizing problems, prosocial behaviour, behavioural self-regulation, cognitive self-regulation and emotional self-regulation. This was inspected using multiple regressions, modelling scores on each CSBQ subscale as a function of age, sex, parental education, cognitive ability and average accuracy on emotional prosody recognition. Results are detailed in table 2. Associations were particularly clear for prosocial behaviour, cognitive self-regulation and behavioural self-regulation, all supported by substantial evidence ($p_s < 0.02$, $3.34 < BF_{10} < 7.78$). We calculated Cook's values and confirmed that the effects were not explained by extreme data points on the regression model: Cook's distance $M = 0.01$, s.d. $= 0.01$ (Cook's distance range $= 0.00–0.06$ for prosocial behaviour; $0.00–0.05$ for behavioural self-regulation; and $0.00–0.06$ for cognitive self-regulation). Partial associations between emotional prosody recognition and these dimensions of socio-emotional adjustment are illustrated in figure 3b–d.

There were also significant associations between emotional prosody recognition and the dimensions of sociability and emotional self-regulation, but the level of evidence was weaker ($p_s < 0.03$, $1.61 < BF_{10} < 2.74$). For the remaining two socio-emotional dimensions, externalizing and internalizing problems, emotional prosody recognition did not uniquely contribute to the models ($p_s > 0.33$, $BF_{10} < 0.18$).

# 4. Discussion

In the current study, we asked whether individual differences in vocal emotion recognition relate to socio-emotional adjustment in children. We measured emotion recognition in two types of vocal emotions, speech prosody and non-verbal vocalizations. Socio-emotional adjustment was assessed through a multidimensional measure completed by the children's teachers. We found strong evidence for a positive association between speech prosody recognition and socio-emotional adjustment, based on both frequentist and Bayesian statistics. This association remained significant even after accounting for age, sex, parental education and cognitive ability. Follow-up analyses showed that prosody recognition was more robustly linked to the socio-emotional dimensions of prosocial behaviour, cognitive self-regulation and behavioural self-regulation. For emotion recognition in non-verbal vocalizations, there were no associations with socio-emotional adjustment. A similar null result was found for the additional emotion recognition task focused on facial expressions.

Some prior studies have reported an association between children's emotional prosody recognition abilities and aspects of socio-emotional adjustment including behavioural problems (e.g. social avoidance and distress; [43]), peer popularity (e.g. [41]) and global social competence (e.g. [44]). However, results have been mixed [41,42] and often based on relatively small samples. It also remained unclear whether the associations are specific, or a result of factors such as parental education. The present study corroborates the association between emotional prosody recognition and socio-emotional adjustment in a sample of 6- to 8-year-olds, and it indicates that this association is not reducible to cognitive or socio-demographic variables, namely age, sex, cognitive ability and parental education. Emotional prosody cues help us build up a mental representation of other's emotional states [3], and prosody can convey a wide range of complex and nuanced states, such as verbal irony, sarcasm and confidence [20,74,75]. Interpreting prosodic cues might be challenging, as indicated by evidence (that we replicated) that emotion recognition accuracy is lower for emotional prosody compared with non-verbal vocalizations and facial expressions (e.g. [13–15]). This increased difficulty might be because prosodic cues are embedded in speech, which constrains acoustic variability [8]. These stimuli are also more complex in that they include both lexico-semantic and prosodic cues, while in non-verbal vocalizations and facial expressions lexico-semantic information is not present. Children with an earlier and more efficient development of this complex ability might therefore be particularly well-equipped to navigate their social worlds.

**Table 2.** Multiple regression analyses for each dimension of socio-emotional adjustment. Predictors were age, sex, parental education, cognitive ability and emotional prosody recognition accuracy. Note: SR, self-regulation. $*p < 0.05$; $**p < 0.01$; $***p < 0.001$ (uncorrected $p$-values).

| model | adj. $R^2$ | $F_{5,133}$ | $BF_{10}$ | $b^a$ | SE | $B^b$ | $t$ | CI 95% | partial $r$ | $BF_{10}$ partial $r$ |
|---|---|---|---|---|---|---|---|---|---|---|
| sociability | 0.19 | 7.46*** | >100 | | | | | | | |
| constant | | | | 5.76 | 0.86 | | 6.73*** | [4.07, 7.46] | | |
| age | | | | −0.43 | 0.11 | −0.31 | −3.97*** | [−0.65, −0.22] | −0.33 | >100 |
| sex | | | | 0.03 | 0.11 | 0.02 | 0.27 | [−0.19, 0.25] | 0.02 | 0.11 |
| parental education | | | | 0.04 | 0.02 | 0.18 | 2.08* | [0.00, 0.07] | 0.18 | 0.91 |
| cognitive ability | | | | 0.01 | 0.01 | 0.06 | 0.69 | [−0.02, 0.04] | 0.06 | 0.14 |
| emotional prosody | | | | 0.76 | 0.33 | 0.19 | 2.34* | [0.12, 1.40] | 0.20 | 1.62 |
| externalizing problems | 0.08 | 3.48** | 2.64 | | | | | | | |
| constant | | | | 0.96 | 0.92 | | 1.04 | [−0.86, 2.78] | | |
| age | | | | 0.21 | 0.12 | 0.15 | 1.79 | [−0.02, 0.44] | 0.15 | 0.53 |
| sex | | | | −0.37 | 0.12 | −0.26 | −3.12** | [−0.60, −0.13] | −0.26 | 12.34 |
| parental education | | | | −0.02 | 0.02 | −0.08 | −0.86 | [−0.05, 0.02] | −0.07 | 0.15 |
| cognitive ability | | | | 0.01 | 0.01 | 0.06 | 0.68 | [−0.02, 0.04] | 0.06 | 0.13 |
| emotional prosody | | | | −0.28 | 0.35 | −0.07 | −0.79 | [−0.97, 0.41] | −0.07 | 0.15 |
| internalizing problems | 0.19 | 7.52*** | >100 | | | | | | | |
| constant | | | | −0.78 | 0.78 | | −1.00 | [−2.33, 0.77] | | |
| age | | | | 0.46 | 0.10 | 0.37 | 4.66*** | [0.27, 0.66] | 0.38 | >100 |
| sex | | | | 0.03 | 0.10 | 0.02 | 0.27 | [−0.17, 0.22] | 0.02 | 0.11 |
| parental education | | | | −0.02 | 0.02 | −0.12 | −1.46 | [−0.05, 0.01] | −0.13 | 0.31 |
| cognitive ability | | | | −0.02 | 0.01 | −0.17 | −1.99* | [−0.05, 0.00] | −0.17 | 0.77 |
| emotional prosody | | | | −0.28 | 0.30 | −0.08 | −0.95 | [−0.87, 0.30] | −0.08 | 0.17 |
| prosocial behaviour | 0.19 | 7.31*** | >100 | | | | | | | |
| constant | | | | 3.62 | 1.86 | | 4.23*** | [1.93, 5.31] | | |
| age | | | | −0.19 | 0.11 | −0.14 | −1.73 | [−0.40, 0.03] | −0.15 | 0.48 |
| sex | | | | 0.27 | 0.11 | 0.19 | 2.46* | [0.05, 0.48] | 0.21 | 2.16 |

(*Continued.*)

**Table 2.** (*Continued.*)

| model | adj. $R^2$ | $F_{5,133}$ | $BF_{10}$ | $b^a$ | SE | $B^b$ | $t$ | CI 95% | partial $r$ | $BF_{10}$ partial $r$ |
|---|---|---|---|---|---|---|---|---|---|---|
| parental education | | | | 0.05 | 0.02 | 0.24 | 2.79** | [0.01, 0.08] | 0.24 | 4.93 |
| cognitive ability | | | | 0.00 | 0.01 | 0.02 | 0.20 | [−0.02, 0.03] | 0.02 | 0.11 |
| emotional prosody | | | | 0.86 | 0.32 | 0.22 | 2.66** | [0.22, 1.51] | 0.23 | 3.51 |
| behavioural SR | 0.19 | 7.56*** | >100 | | | | | | | |
| constant | | | | 3.00 | 0.99 | | 3.03** | [1.04, 4.96] | | |
| age | | | | −0.14 | 0.13 | −0.09 | −1.14 | [−0.39, 0.11] | −0.10 | 0.21 |
| sex | | | | 0.40 | 0.13 | 0.25 | 3.21** | [0.16, 0.65] | 0.27 | 16.34 |
| parental education | | | | 0.06 | 0.02 | 0.24 | 2.87** | [0.02, 0.10] | 0.24 | 6.11 |
| cognitive ability | | | | −0.00 | 0.02 | −0.02 | −0.21 | [−0.03, 0.03] | −0.02 | 0.11 |
| emotional prosody | | | | 0.99 | 0.38 | 0.21 | 2.64** | [0.25, 1.74] | 0.22 | 3.35 |
| cognitive SR | 0.43 | 21.47*** | >100 | | | | | | | |
| constant | | | | 2.59 | 1.02 | | 2.54* | [0.57, 4.62] | | |
| age | | | | −0.26 | 0.13 | −0.18 | −2.78** | [−0.62, −0.10] | −0.23 | 4.76 |
| sex | | | | 0.08 | 0.13 | 0.04 | 0.61 | [−0.18, 0.34] | 0.05 | 0.13 |
| parental education | | | | 0.11 | 0.02 | 0.37 | 5.13*** | [0.07, 0.15] | 0.41 | >100 |
| cognitive ability | | | | 0.06 | 0.02 | 0.27 | 3.82*** | [0.03, 0.09] | 0.31 | >100 |
| emotional prosody | | | | 1.15 | 0.39 | 0.20 | 2.96** | [0.38, 1.91] | 0.25 | 7.77 |
| emotional SR | 0.09 | 3.82** | 5.05 | | | | | | | |
| constant | | | | 4.55 | 0.92 | | 4.92** | [2.72, 6.37] | | |
| age | | | | −0.17 | 0.12 | −0.12 | −1.43 | [−0.40, 0.06] | −0.12 | 0.30 |
| sex | | | | 0.32 | 0.12 | 0.23 | 2.74** | [0.09, 0.55] | 0.23 | 4.34 |
| parental education | | | | 0.01 | 0.02 | 0.06 | 0.61 | [−0.03, 0.05] | 0.05 | 0.13 |
| cognitive ability | | | | −0.02 | 0.01 | −0.13 | −1.42 | [−0.05, 0.01] | −0.12 | 0.29 |
| emotional prosody | | | | 0.90 | 0.35 | 0.22 | 2.56* | [0.20, 1.59] | 0.22 | 2.73 |

[a]Unstandardized regression coefficient.

[b]Standardized regression coefficient.

In exploratory analyses focused on specific dimensions of socio-emotional adjustment, we found that children's ability to recognize emotional prosody was particularly related to prosocial behaviour and cognitive and behavioural self-regulation. These findings were based on uncorrected $p$-values, but the fact that they were also supported by substantial Bayesian evidence suggests that they are meaningful. Prosociality is associated with positive social behaviours such as cooperation, altruism and empathy [76,77]. The ability to recognize fearful facial expressions was found to be linked to adults' prosocial behaviour [78–80]. This could be because distress cues are a powerful tool to elicit care, and being able to 'read' them could promote prosocial behaviours, such as helping a crying child [81]. Regarding vocal emotions, decreased cooperative behaviour was observed in adults towards partners displaying emotional prosody of anger, fear and disgust [82]. However, this was found in a study focused on decisions to cooperate in a social decision-making paradigm, and participants' ability to recognize emotional prosody was not examined. To our knowledge, the current study is the first to show that emotional prosody recognition is positively linked to prosocial behaviour in school-aged children. It is possible that the ability to accurately interpret the emotional meaning of complex stimuli (such as speech) allows children to more readily deduce when to cooperate, share or help others, all prosocial behaviours covered by our measure. Future work inspecting how children's vocal emotion recognition relates to their prosocial behaviour will be important to better understand this finding.

Self-regulation includes behavioural and cognitive components, and we found associations with children's prosody recognition abilities for both. The behavioural component refers to the ability to remain on task, to inhibit behaviours that might not contribute to goal achievement, and to follow socially appropriate rules [26]. The cognitive component is focused on more top-down processes related to problem-solving, focused attention and self-monitoring, which might support autonomy and task persistence. Prior evidence shows that pre-schoolers' recognition of facial expressions correlates with attention processes and behavioural self-regulation [55,56], but evidence regarding vocal emotion recognition is scant. In view of evidence that attention can contribute to performance in emotional prosody tasks in adults (e.g. [31,83]) and children (e.g. [84]), it could have been that children who were more able to focus and remain on task were in a better position for improved performance. For instance, emotional prosody recognition requires listeners to maintain temporally dynamic information in working memory to inform interpretation, and self-regulation may covary with this type of attention [85]. However, although we found a correlation between cognitive ability and prosody recognition, thus replicating previous evidence, the association with self-regulation remained significant after cognitive ability was accounted for, making this explanation less likely. Alternatively, because the ability to decode emotional prosody supports a more efficient understanding of communicative messages (e.g. from parents or teachers), this might allow children to understand more easily the tasks they are expected to perform, the rules to follow and the goals to achieve. Future studies assessing self-regulatory processes in more detail will be important to delineate the sub-processes driving the general associations uncovered here.

Contrasting with the findings for prosody, for non-verbal vocalizations, we observed no associations with socio-emotional adjustment. To our knowledge, ours is the first study that systematically considers the two sources of vocal emotional cues—prosody and non-verbal vocalizations—in the context of associations with socio-emotional functioning. This matters because, despite both being vocal emotional expressions, they differ in their production and perceptual mechanisms [7,8], and indeed also seem to differ in their correlates. This null result seems unexpected, considering that non-verbal vocalizations reflect a primitive and universal form of communication (e.g. [12]), thought to play an important role in social interactions. It could have been that our measures of emotion recognition and socio-emotional adjustment were not sensitive enough to capture the effect. But it could also be that variability in the processing of vocalizations does not play a major role for socio-emotional functioning in typically developing school-age children. Previous results indicate that children as young as 5 years are already proficient at recognizing a range of positive and negative emotions in non-verbal vocalizations, with average accuracy approaching 80%, and there is no improvement from 5 to 10 years for most emotions [15]. Such proficiency is replicated here, and we also found that the range of individual differences is small when compared with prosody (figure 1). This could mean that, for most healthy school-age children, the ability to recognize non-verbal emotional vocalizations is already high enough for them to optimally use these cues in social interactions, such that small individual variation will not necessarily translate into measurable differences in everyday behaviour. This result will need to be followed up in future studies, however, to examine whether it replicates across different measures and age groups (e.g. including a broader range of emotions and a more comprehensive assessment of socio-emotional adjustment).

That performance on the additional facial emotion recognition task also did not correlate with socio-emotional adjustment corroborates the findings of some previous studies. McClure & Nowicki [43] found that 8- to 10-year-old children's ability to recognize facial expressions was not associated with dimensions of socio-emotional adjustment, namely social avoidance and distress. Leppänen & Hietanen [44] also reported null results regarding peer popularity in a sample of 7- to 10-year-olds. Moreover, Chronaki *et al.* [42] found that pre-schoolers' ability to recognize facial expressions was not associated with parent-rated internalizing problems. On the other hand, there is evidence that facial emotion recognition can relate to fewer behavioural problems in school-age children (e.g. [46]) and to better self-regulation in pre-schoolers (e.g. [56]). These discrepancies across studies might stem from differences in samples' characteristics and measures. For instance, pre-schoolers [56] compared with school-age children [43], and measures of peer-rated popularity [44] compared with measures of social avoidance and distress [43]. Such possibilities will be clarified as more research is conducted on this topic. In the current study, based on a relatively large sample informed by power analyses, Bayesian statistics provided in fact evidence for the null hypothesis. In line with our reasoning for non-verbal vocalizations, a tentative explanation is that children's proficiency at decoding facial emotions at this age is already high, such that the impact of individual variation in everyday life behaviour might be less apparent.

A limitation of the current study is the correlational approach. We provide evidence for an association between emotional prosody recognition and socio-emotional adjustment, but we cannot exclude the possibility that emotional prosody recognition skills are the result, not the cause, of better socio-emotional adjustment. Having more and better social interactions plausibly provides opportunities for children to learn about emotional expressions and to hone their emotion recognition skills. Future systematic longitudinal research will be needed to establish causality, for example by testing whether an emotion recognition training programme leads to improved social interactions. Another limitation is that we used vocal and facial stimuli produced by adults, and it would be interesting to know if similar results would be obtained with stimuli produced by children. Children can accurately recognize vocal expressions produced by participants of any age, but there is also evidence that they might perform better for stimuli produced by children their age ([18,86]; but see [43]). Moreover, the emotional prosody task contained stimuli produced by female speakers only, whereas non-verbal vocalizations and facial expressions included both female and male actors. Because there is some previous evidence that the speaker's sex might influence vocal emotion recognition (e.g. [87,88]; but see [18]), we cannot exclude the possibility this might have contributed to the distinct results across tasks. Future studies should also extend our findings to different emotion recognition tasks to establish their generalizability. In line with previous studies (e.g. [15,18,60]), we have used visual aids (emojis) to make the task more engaging and less reliant on linguistic/reading abilities, but at the same time, this might have inflated performance and increased the reliance on auditory-visual matching processes. One last point is that we only used a teacher report socio-emotional measure. Future work combining different socio-emotional measures, such as parent report and performance-based tasks, would allow us to test these relationships more stringently.

In conclusion, the current study shows that emotional speech prosody recognition is associated with general socio-emotional adjustment in children. We also show that this association is not explained by cognitive and socio-demographic variables, and results were particularly robust for the socio-emotional dimensions of prosocial behaviour and self-regulation (cognitive and behavioural components). These findings did not generalize to vocal emotional stimuli without linguistic information—non-verbal vocalizations—and were also not seen for facial expressions. Altogether, these results support the notion that emotional speech recognition skills play an important role in children's everyday social interactions. They also contribute to debates on the functional role of vocal emotional expressions and might inform interventions aimed at fostering socio-emotional skills in childhood.

Research ethics. Ethical approval for the study protocol was obtained from the local Ethics Committee, ISCTE-IUL (reference 28/2019). Written informed consent was collected from all participants from a parent or legal guardian, and children gave verbal assent to participate.

Data accessibility. The dataset supporting this article is freely available for public use at the OSF platform: https://osf.io/qfp83/?view_only=47031990843a48978ca8058e98118805.

Authors' contributions. L.N. and M.M. participated in the design of the study, prepared the tasks, collected the data, conducted data analysis and interpretation, and drafted the manuscript. A.I.C. collected the data, participated in data analysis and helped draft the manuscript. S.L.C. helped conceive and design the study, coordinated the study,

and helped draft the manuscript. C.F.L. conceived and designed the study, coordinated the study, participated in data analysis and interpretation, and drafted the manuscript. All authors gave final approval for publication.

Competing interests. C.F.L. was a member of the Royal Society Open Science editorial board at the time of submission; however, they were not involved in the editorial assessment of the manuscript in any way.

Funding. This work was funded by the Portuguese Foundation for Science and Technology (FCT) through a PhD studentship awarded to L.N. (SFRH/BD/135604/2018), a FCT Investigator grant awarded to C.F.L (IF/00172/2015), and a project grant awarded to C.F.L. (PTDC/PSI-GER/28274/2017) and co-funded by the European Regional Development Fund (ERDF) through the Lisbon Regional Operational Program (LISBOA-01-0145-FEDER-028274) and the Operational Program for Competitiveness and Internationalization (POCI-01-0145-FEDER-028274).

Acknowledgements. We thank the school administration, teachers, parents and all the children who took part in the study.

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
