## [Peer Review File · Royal Society Open Science]

Review History

RSOS-210454.R0 (Original submission)

Review form: Reviewer 1

Is the manuscript scientifically sound in its present form?

Yes

Are the interpretations and conclusions justified by the results?

Yes

Is the language acceptable?

Yes

Do you have any ethical concerns with this paper?

No

Have you any concerns about statistical analyses in this paper?

No

Recommendation?

Major revision is needed (please make suggestions in comments)

Comments to the Author(s)

This is an interesting study which has examined the important topic of the association between emotion recognition in speech prosody and socio-emotional adjustment in 6-8-year old children. The study has followed a correlational design and has found that higher emotion recognition in speech prosody related to better general socio-emotional adjustment in children as rated by teachers. A strength of this study is the large sample size. However, the present study has a number of theoretical and conceptual limitations which, in its current form, prohibit publication in this journal. I have offered some suggestions below which I hope the authors will find helpful in improving the manuscript.

1. The study found that higher emotion recognition in speech prosody related to better general socio-emotional adjustment. Associations between recognition of vocal emotional expressions and social competence have been examined by previous research in pre-schoolers and school-aged children (see for example studies by Verbeek, 1996, Mitchell, 1995, Baum & Nowicki, 1998, Rothman & Nowicki, 2004, Goonan, 1995, see also Nowicki & Mitchell, 1998 at Genetic Social and General Psychology Monographs etc.). It is not clear what is new information here and what the present study adds to the current knowledge base. It needs to be very clearly explained in the introduction what exactly the present study adds to each previous study that has examined this topic.
2. In the abstract the following sentence seems to contradict the previous sentence stating that this association was found. 'For emotion recognition in nonverbal vocalisations, no associations with socio-emotional adjustment were found'.
3. In the introduction (see first page) the authors state 'But it has also been shown that emotion recognition accuracy is higher for nonverbal vocalisations than for emotional prosody'. Please, can the authors explain the distinction between 'nonverbal vocalisations' and 'emotional prosody' as these terms read like they refer to the same construct, nonverbal vocalisations refer to emotional prosody as opposed to verbal vocalisations.
4. I would avoid extensive mention to clinical populations in the introduction if the present study only recruits children from the general population. I understand why the authors cite studies by Nowicki and colleagues and Chronaki et al. as these studies have focused on children from the general population, but it is not clear why the introduction discuss clinical samples (see page 4) without this being the focus of the present study.
5. Can the authors offer some background information in the introduction about why they expect the association between speech prosody related to better general socio-emotional adjustment to remain significant even when individual differences in age, sex, parental education, and cognitive ability are accounted for. This hypothesis comes as a surprise the reader without any theoretical context in the introduction. The same applies to the 'children completed an additional emotion recognition task focused on facial expressions'. This sentence also comes as a surprise in the last sentences of the introduction as the whole introduction so far has focused on vocal expressions.
6. In the methods section the authors state that to improve children's engagement throughout the task, an emoji illustrating each emotional category was included both on the response pad and on the laptop scree. Have the authors considered that matching the vocal emotional expression with an emotional facial expression (emoji) may be a confounding variable in recognition?
7. Regarding the methods, could the authors explain the theoretical rationale for which socio-emotional adjustment (assessed via the CSBQ questionnaire) was completed by the children's teacher only.
8. Please, could you include some brief description in text of the associations between specific emotions and socio-emotional adjustment at the bottom of page 16 rather than only including these results in the table.

9. Page 17 of the results section is not accessible for the reader and needs to be condensed considerably. Please only include a small paragraph with the most important/strongest associations as the key message and include all other statistical details in a table.

10. The results of page 17 are difficult to interpret in the absence of a theoretical rationale in the introduction linking specific sub-scales/dimensions of the socio-emotional adjustment questionnaire measure (e.g., prosocial behaviour, behavioural self-regulation, cognitive self-regulation) to vocal emotion recognition. It is not clear why these associations are important and what they mean. Some conceptual framework is necessary in order to understand this.

11. The discussion needs more work to add some clarity on the issues of theoretical contribution of the present study to the existing knowledge base. Also there are some issues with terminology (e.g. prosody typically refers to nonverbal vocalisations in the literature as opposed to verbal vocalisations) - these issues also need to be examined more carefully.

Review form: Reviewer 2

Is the manuscript scientifically sound in its present form?

Yes

Are the interpretations and conclusions justified by the results?

Yes

Is the language acceptable?

Yes

Do you have any ethical concerns with this paper?

No

Have you any concerns about statistical analyses in this paper?

Yes

Recommendation?

Major revision is needed (please make suggestions in comments)

Comments to the Author(s)

This study examines the association between socio-emotional functioning and emotion recognition abilities for prosody, vocalizations, and facial expressions in 6- to 8-year-old children. The authors report that greater socio-emotional adjustment was associated with better recognition of prosody (even after controlling for cognitive ability, age, sex, and parental education), but not of other forms of nonverbal cues. I very much enjoyed reading this paper. The manuscript is very well written and clear. The approach was thorough and sound, and I appreciated the thoughtful discussion of the findings. I have a few questions and suggestions to improve clarity in some parts, and to recommend consideration of alternative interpretations of the findings. Details are below.

Introduction

- The authors present evidence about the association between vocal emotion recognition and psychosocial functioning derived from adult clinical populations. However, they omit relevant evidence with pediatric clinical populations (e.g., Manassis & Young, 2000; Emerson, Harrison, & Everhart, 1999; Morningstar et al., 2019; Deveney et al., 2012). Given the age range of interest, it seems more pertinent to refer to this literature than to work with schizophrenia or Parkinson's, for instance.

Methods

- Hu values should be arcsine-transformed before analyses (Wagner, 1993). Was this step performed?
- Although averaging across emotions to obtain one Hu score is adequate, many modern studies examine whether there are emotion-specific associations with variables of interest. Given that the evidence the authors review in the introduction and discussion includes emotion-specific links between ER performance and clinical/psychosocial variables, a similar analysis would serve to better contextualize findings within the existing literature. The authors do perform separate models for each emotion type, but this approach ignores the interdependence of ER scores across emotion types. A more thorough investigation of potential emotion-specific links with socio-emotional functioning may be to consider the different emotion types as repeated-measures within a general linear model (which would still allow the authors to include covariates of interest).
- More details about the analytical models are needed, particularly regarding the frequentist analyses. What factors were included into the models and how were they operationalized? For instance, when the authors indicate that “performance differed significantly across tasks, $F...$ ”, I assume an ANOVA was performed – but I cannot tell whether factors other than ‘modality’ were entered into the model or controlled for. Including more information about the planned statistical analyses would ensure readers do not have to make assumptions.
- Minor point: The prosody ER task contained stimuli spoken by female speakers only, whereas the vocalization and facial expression ER tasks included both male and female encoders. Given that speaker gender has been found to influence the recognizability of emotional prosody, could the authors speculate as to whether this aspect of the design could have contributed to differential patterns of results pertaining to the prosody vs. the other two tasks – or note this as a limitation, perhaps?

Results

- I am confused about the application of Holm-Bonferroni corrections or about how it is reported. In the Tables, do asterisks mean that an effect was still considered significant after Holm-Bonferroni was applied? And then, in Table S5, when there are no asterisks... does this mean that none of these p-values were above the corrected Holm-Bonferroni threshold? It is difficult to tell for sure without the p-values being reported.
- Hu is lower than I would expect for sadness (in prosody), which was recognized at the same level as disgust in the current study. Typically, sadness is well-recognized in emotional prosody. An inspection of the confusion matrices may be warranted to better understand the source of unexpectedly low accuracy for this emotion.
- Minor point: some of the text in p. 17 could be put into a Table to make it easier for readers to parse this section and compare the contribution of variables of interest across subscales.

Discussion

- Were children screened for autism spectrum disorders? Autism symptoms could be an important ‘third variable’ to consider in interpreting findings, given its known association with deficits in vocal ER and socio-emotional functioning.
- A potential interpretation of the link between behavioural and cognitive SR and prosody ER may be that, compared to vocalizations and still images of faces, emotional prosody requires listeners to ‘hold’ temporally dynamic information in working memory across longer periods of time to inform interpretation. Self-regulation may thus covary with the type of attention and executive functions that would facilitate the interpretation of prosody, but may not be required as much for other forms of nonverbal cues.
- Minor point: The authors argue that “discrepancies across studies might stem from differences in samples’ characteristics and measures” – such as? Though the authors would need to speculate, extending this discussion could help direct future research.

Decision letter (RSOS-210454.R0)

Dear Dr Lima

The Editors assigned to your paper RSOS-210454 "Associations Between Vocal Emotion Recognition and Socio-emotional Adjustment in Children" have made a decision based on their reading of the paper and any comments received from reviewers.

Regrettably, in view of the reports received, the manuscript has been rejected in its current form. However, a new manuscript may be submitted which takes into consideration these comments.

We invite you to respond to the comments supplied below and prepare a resubmission of your manuscript. Below the referees' and Editors' comments (where applicable) we provide additional requirements. We provide guidance below to help you prepare your revision.

Please note that resubmitting your manuscript does not guarantee eventual acceptance, and we do not generally allow multiple rounds of revision and resubmission, so we urge you to make every effort to fully address all of the comments at this stage. If deemed necessary by the Editors, your manuscript will be sent back to one or more of the original reviewers for assessment. If the original reviewers are not available, we may invite new reviewers.

Please resubmit your revised manuscript and required files (see below) no later than 19-Dec-2021. Note: the ScholarOne system will 'lock' if resubmission is attempted on or after this deadline. If you do not think you will be able to meet this deadline, please contact the editorial office immediately.

Please note article processing charges apply to papers accepted for publication in Royal Society Open Science (<https://royalsocietypublishing.org/rsos/charges>). Charges will also apply to papers transferred to the journal from other Royal Society Publishing journals, as well as papers submitted as part of our collaboration with the Royal Society of Chemistry (<https://royalsocietypublishing.org/rsos/chemistry>). Fee waivers are available but must be requested when you submit your manuscript (<https://royalsocietypublishing.org/rsos/waivers>).

Thank you for submitting your manuscript to Royal Society Open Science and we look forward to receiving your resubmission. If you have any questions at all, please do not hesitate to get in touch.

on behalf of Dr Teodora Gliga (Associate Editor) and Essi Viding (Subject Editor)
openscience@royalsociety.org

Associate Editor Comments to Author (Dr Teodora Gliga):

Associate Editor: 1

Comments to the Author:

I have now received comments from 2 reviewers which carefully assessed your manuscript. As you will see, they both see some merit in your contribution but also raise major concerns with respect to 1) the statistical analyses employed (both reviewers) and 2) the presence of some confounds (rev 1). I therefore decide to reject this version of the manuscript but allow a resubmission. Apart from the two points highlighted above I also urge you to make a more compelling case for what your study adds to the existing literature as well as to improve the clarity of the statistical reporting (as per reviewer's comments below).

Reviewer comments to Author:

Reviewer: 1

Comments to the Author(s)

This is an interesting study which has examined the important topic of the association between emotion recognition in speech prosody and socio-emotional adjustment in 6-8-year old children. The study has followed a correlational design and has found that higher emotion recognition in speech prosody related to better general socio-emotional adjustment in children as rated by teachers. A strength of this study is the large sample size. However, the present study has a number of theoretical and conceptual limitations which, in its current form, prohibit publication in this journal. I have offered some suggestions below which I hope the authors will find helpful in improving the manuscript.

1. The study found that higher emotion recognition in speech prosody related to better general socio-emotional adjustment. Associations between recognition of vocal emotional expressions and social competence have been examined by previous research in pre-schoolers and school-aged children (see for example studies by Verbeek, 1996, Mitchell, 1995, Baum & Nowicki, 1998, Rothman & Nowicki, 2004, Goonan, 1995, see also Nowicki & Mitchell, 1998 at Genetic Social and General Psychology Monographs etc.). It is not clear what is new information here and what the present study adds to the current knowledge base. It needs to be very clearly explained in the introduction what exactly the present study adds to each previous study that has examined this topic.
2. In the abstract the following sentence seems to contradict the previous sentence stating that this association was found. 'For emotion recognition in nonverbal vocalisations, no associations with socio-emotional adjustment were found'.
3. In the introduction (see first page) the authors state 'But it has also been shown that emotion recognition accuracy is higher for nonverbal vocalisations than for emotional prosody'. Please, can the authors explain the distinction between 'nonverbal vocalisations' and 'emotional prosody' as these terms read like they refer to the same construct, nonverbal vocalisations refer to emotional prosody as opposed to verbal vocalisations.
4. I would avoid extensive mention to clinical populations in the introduction if the present study only recruits children from the general population. I understand why the authors cite studies by Nowicki and colleagues and Chronaki et al. as these studies have focused on children from the general population, but it is not clear why the introduction discuss clinical samples (see page 4) without this being the focus of the present study.
5. Can the authors offer some background information in the introduction about why they expect the association between speech prosody related to better general socio-emotional adjustment to remain significant even when individual differences in age, sex, parental education, and cognitive ability are accounted for. This hypothesis comes as a surprise the reader without any theoretical context in the introduction. The same applies to the 'children completed an additional emotion recognition task focused on facial expressions'. This sentence also comes as a surprise in the last sentences of the introduction as the whole introduction so far has focused on vocal expressions.

6. In the methods section the authors state that to improve children's engagement throughout the task, an emoji illustrating each emotional category was included both on the response pad and on the laptop screen. Have the authors considered that matching the vocal emotional expression with an emotional facial expression (emoji) may be a confounding variable in recognition?
7. Regarding the methods, could the authors explain the theoretical rationale for which socio-emotional adjustment (assessed via the CSBQ questionnaire) was completed by the children's teacher only.
8. Please, could you include some brief description in text of the associations between specific emotions and socio-emotional adjustment at the bottom of page 16 rather than only including these results in the table.
9. Page 17 of the results section is not accessible for the reader and needs to be condensed considerably. Please only include a small paragraph with the most important/strongest associations as the key message and include all other statistical details in a table.
10. The results of page 17 are difficult to interpret in the absence of a theoretical rationale in the introduction linking specific sub-scales/ dimensions of the socio-emotional adjustment questionnaire measure (e.g., prosocial behaviour, behavioural self-regulation, cognitive self-regulation) to vocal emotion recognition. It is not clear why these associations are important and what they mean. Some conceptual framework is necessary in order to understand this.
11. The discussion needs more work to add some clarity on the issues of theoretical contribution of the present study to the existing knowledge base. Also there are some issues with terminology (e.g. prosody typically refers to nonverbal vocalisations in the literature as opposed to verbal vocalisations) - these issues also need to be examined more carefully.

Reviewer: 2

Comments to the Author(s)

This study examines the association between socio-emotional functioning and emotion recognition abilities for prosody, vocalizations, and facial expressions in 6- to 8-year-old children. The authors report that greater socio-emotional adjustment was associated with better recognition of prosody (even after controlling for cognitive ability, age, sex, and parental education), but not of other forms of nonverbal cues. I very much enjoyed reading this paper. The manuscript is very well written and clear. The approach was thorough and sound, and I appreciated the thoughtful discussion of the findings. I have a few questions and suggestions to improve clarity in some parts, and to recommend consideration of alternative interpretations of the findings. Details are below.

Introduction

- The authors present evidence about the association between vocal emotion recognition and psychosocial functioning derived from adult clinical populations. However, they omit relevant evidence with pediatric clinical populations (e.g., Manassis & Young, 2000; Emerson, Harrison, & Everhart, 1999; Morningstar et al., 2019; Deveney et al., 2012). Given the age range of interest, it seems more pertinent to refer to this literature than to work with schizophrenia or Parkinson's, for instance.

Methods

- Hu values should be arcsine-transformed before analyses (Wagner, 1993). Was this step performed?

- Although averaging across emotions to obtain one Hu score is adequate, many modern studies examine whether there are emotion-specific associations with variables of interest. Given that the evidence the authors review in the introduction and discussion includes emotion-specific links between ER performance and clinical/psychosocial variables, a similar analysis would serve to better contextualize findings within the existing literature. The authors do perform separate models for each emotion type, but this approach ignores the interdependence of ER scores across

emotion types. A more thorough investigation of potential emotion-specific links with socio-emotional functioning may be to consider the different emotion types as repeated-measures within a general linear model (which would still allow the authors to include covariates of interest).

- More details about the analytical models are needed, particularly regarding the frequentist analyses. What factors were included into the models and how were they operationalized? For instance, when the authors indicate that “performance differed significantly across tasks, $F...$ ”, I assume an ANOVA was performed – but I cannot tell whether factors other than ‘modality’ were entered into the model or controlled for. Including more information about the planned statistical analyses would ensure readers do not have to make assumptions.

- Minor point: The prosody ER task contained stimuli spoken by female speakers only, whereas the vocalization and facial expression ER tasks included both male and female encoders. Given that speaker gender has been found to influence the recognizability of emotional prosody, could the authors speculate as to whether this aspect of the design could have contributed to differential patterns of results pertaining to the prosody vs. the other two tasks – or note this as a limitation, perhaps?

Results

- I am confused about the application of Holm-Bonferroni corrections or about how it is reported. In the Tables, do asterisks mean that an effect was still considered significant after Holm-Bonferroni was applied? And then, in Table S5, when there are no asterisks... does this mean that none of these p-values were above the corrected Holm-Bonferroni threshold? It is difficult to tell for sure without the p-values being reported.

- Hu is lower than I would expect for sadness (in prosody), which was recognized at the same level as disgust in the current study. Typically, sadness is well-recognized in emotional prosody. An inspection of the confusion matrices may be warranted to better understand the source of unexpectedly low accuracy for this emotion.

- Minor point: some of the text in p. 17 could be put into a Table to make it easier for readers to parse this section and compare the contribution of variables of interest across subscales.

Discussion

- Were children screened for autism spectrum disorders? Autism symptoms could be an important ‘third variable’ to consider in interpreting findings, given its known association with deficits in vocal ER and socio-emotional functioning.

- A potential interpretation of the link between behavioural and cognitive SR and prosody ER may be that, compared to vocalizations and still images of faces, emotional prosody requires listeners to ‘hold’ temporally dynamic information in working memory across longer periods of time to inform interpretation. Self-regulation may thus covary with the type of attention and executive functions that would facilitate the interpretation of prosody, but may not be required as much for other forms of nonverbal cues.

- Minor point: The authors argue that “discrepancies across studies might stem from differences in samples’ characteristics and measures” – such as? Though the authors would need to speculate, extending this discussion could help direct future research.

===PREPARING YOUR MANUSCRIPT===

a ‘clean’ version of the new manuscript that incorporates the changes made, but does not highlight them. This version will be used for typesetting if your manuscript is accepted.

===PREPARING YOUR REVISION IN SCHOLARONE===

- If you are providing image files for potential cover images, please upload these at this step, and inform the editorial office you have done so. You must hold the copyright to any image provided.
- A copy of your point-by-point response to referees and Editors. This will expedite the preparation of your proof.

- Ensure that your data access statement meets the requirements at <https://royalsociety.org/journals/authors/author-guidelines/#data>. You should ensure that you cite the dataset in your reference list. If you have deposited data etc in the Dryad repository, please include both the 'For publication' link and 'For review' link at this stage.
- If you are requesting an article processing charge waiver, you must select the relevant waiver option (if requesting a discretionary waiver, the form should have been uploaded at Step 3 'File upload' above).
- If you have uploaded ESM files, please ensure you follow the guidance at <https://royalsociety.org/journals/authors/author-guidelines/#supplementary-material> to include a suitable title and informative caption. An example of appropriate titling and captioning may be found at [https://figshare.com/articles/Table_S2_from_Is_there_a_trade-off_between_peak_performance_and_performance_breadth_across_temperatures_for_aerobic_sc](https://figshare.com/articles/Table_S2_from_Is_there_a_trade-off_between_peak_performance_and_performance_breadth_across_temperatures_for_aerobic_scope_in_teleost_fishes_/3843624) ope_in_teleost_fishes_/3843624.

Author's Response to Decision Letter for (RSOS-210454.R0)

See Appendix A.

RSOS-211412.R0

Review form: Reviewer 2

Is the manuscript scientifically sound in its present form?

Yes

Are the interpretations and conclusions justified by the results?

Yes

Is the language acceptable?

Yes

Do you have any ethical concerns with this paper?

No

Have you any concerns about statistical analyses in this paper?

No

Recommendation?

Accept with minor revision (please list in comments)

Comments to the Author(s)

The authors have done a great job of addressing my concerns. I have a few remaining suggestions intended to allow further transparency and replicability.

- 1) Although I recognize that interpretation of untransformed Hu values is easier for the reader, arcsine transformations of Hu are recommended (given that Hu values are proportions; Wagner, 1993) and widely applied in the literature. As such, including the results of analyses with transformed values is valuable to researchers seeking to estimate effect sizes, replicate findings, conduct meta-analyses, etc. This could be done in Supplemental Materials.
- 2) The authors indicate that they did not correct p-values obtained in follow-up analyses because these were designed to answer exploratory questions. However, the results from these analyses (particularly in regards to subscales of socio-emotional adjustment) are treated as similar in robustness to the corrected results in the discussion. If the aim is to inform future studies, these exploratory results should be presented transparently as both uncorrected and corrected. The Bayesian statistics help make the claim that these are meaningful findings, but all relevant information should be provided to readers.
- 3) Minor point: The authors include the null findings for facial ER in the discussion and conclusions, but not in the abstract. If there is space, it would be good to present those findings there too (as a further contrast to the prosody findings).

Decision letter (RSOS-211412.R0)

Dear Dr Lima

On behalf of the Editors, we are pleased to inform you that your Manuscript RSOS-211412 "Associations Between Vocal Emotion Recognition and Socio-emotional Adjustment in Children" has been accepted for publication in Royal Society Open Science subject to minor revision in accordance with the referees' reports. Please find the referees' comments along with any feedback from the Editors below my signature.

Please submit your revised manuscript and required files (see below) no later than 7 days from today's (ie 11-Oct-2021) date. Note: the ScholarOne system will 'lock' if submission of the revision is attempted 7 or more days after the deadline. If you do not think you will be able to meet this deadline please contact the editorial office immediately.

on behalf of Dr Teodora Gliga (Associate Editor) and Essi Viding (Subject Editor)
openscience@royalsociety.org

Associate Editor Comments to Author (Dr Teodora Gliga):

Comments to the Author:

Thank you for re-submitting this manuscript. I agree with one of the original reviewers, who has seen your revised manuscript, that this is substantially improved and this will be a nice contribution to the emerging literature on verbal and non-verbal emotional cues. There remain a couple of small issues which you will have to address (see reviewer's comments below).

Reviewer comments to Author:

Reviewer: 2

Comments to the Author(s)

The authors have done a great job of addressing my concerns. I have a few remaining suggestions intended to allow further transparency and replicability.

1) Although I recognize that interpretation of untransformed H_u values is easier for the reader, arcsine transformations of H_u are recommended (given that H_u values are proportions; Wagner, 1993) and widely applied in the literature. As such, including the results of analyses with transformed values is valuable to researchers seeking to estimate effect sizes, replicate findings, conduct meta-analyses, etc. This could be done in Supplemental Materials.

2) The authors indicate that they did not correct p-values obtained in follow-up analyses because these were designed to answer exploratory questions. However, the results from these analyses (particularly in regards to subscales of socio-emotional adjustment) are treated as similar in robustness to the corrected results in the discussion. If the aim is to inform future studies, these exploratory results should be presented transparently as both uncorrected and corrected. The Bayesian statistics help make the claim that these are meaningful findings, but all relevant information should be provided to readers.

3) Minor point: The authors include the null findings for facial ER in the discussion and conclusions, but not in the abstract. If there is space, it would be good to present those findings there too (as a further contrast to the prosody findings).

===PREPARING YOUR MANUSCRIPT===

===PREPARING YOUR REVISION IN SCHOLARONE===

- An individual file of each figure (EPS or print-quality PDF preferred [either format should be produced directly from original creation package], or original software format).
- An editable file of each table (.doc, .docx, .xls, .xlsx, or .csv).
- An editable file of all figure and table captions.

- Any electronic supplementary material (ESM).
- If you are requesting a discretionary waiver for the article processing charge, the waiver form must be included at this step.
- If you are providing image files for potential cover images, please upload these at this step, and inform the editorial office you have done so. You must hold the copyright to any image provided.
- A copy of your point-by-point response to referees and Editors. This will expedite the preparation of your proof.

- Ensure that your data access statement meets the requirements at <https://royalsociety.org/journals/authors/author-guidelines/#data>. You should ensure that you cite the dataset in your reference list. If you have deposited data etc in the Dryad repository, please only include the 'For publication' link at this stage. You should remove the 'For review' link.
- If you are requesting an article processing charge waiver, you must select the relevant waiver option (if requesting a discretionary waiver, the form should have been uploaded at Step 3 'File upload' above).
- If you have uploaded ESM files, please ensure you follow the guidance at <https://royalsociety.org/journals/authors/author-guidelines/#supplementary-material> to include a suitable title and informative caption. An example of appropriate titling and captioning may be found at https://figshare.com/articles/Table_S2_from_Is_there_a_trade-off_between_peak_performance_and_performance_breadth_across_temperatures_for_aerobic_scope_in_teleost_fishes_/3843624.

Author's Response to Decision Letter for (RSOS-211412.R0)

See Appendix B.

Decision letter (RSOS-211412.R1)

Dear Dr Lima,

I am pleased to inform you that your manuscript entitled "Associations Between Vocal Emotion Recognition and Socio-emotional Adjustment in Children" is now accepted for publication in Royal Society Open Science.

on behalf of Dr Teodora Gliga (Associate Editor) and Essi Viding (Subject Editor)
openscience@royalsociety.org

Appendix A

Response to the comments

Associate Editor

I have now received comments from 2 reviewers which carefully assessed your manuscript. As you will see, they both see some merit in your contribution but also raise major concerns with respect to 1) the statistical analyses employed (both reviewers) and 2) the presence of some confounds (rev 1). I therefore decide to reject this version of the manuscript but allow a resubmission. Apart from the two points highlighted above I also urge you to make a more compelling case for what your study adds to the existing literature as well as to improve the clarity of the statistical reporting (as per reviewer's comments below).

We have addressed all the reviewers' concerns about statistical analyses and confounds (please see comment 6, Reviewer 1; and comments 2-4 and 6, Reviewer 2). We have also elaborated on why our study is a novel contribution to the literature (please see comments 1-3, Reviewer 1).

Reviewer 1

This is an interesting study which has examined the important topic of the association between emotion recognition in speech prosody and socio-emotional adjustment in 6-8-year-old children. The study has followed a correlational design and has found that higher emotion recognition in speech prosody related to better general socio-emotional adjustment in children as rated by teachers. A strength of this study is the large sample size. However, the present study has a number of theoretical and conceptual limitations which, in its current form, prohibit publication in this journal. I have offered some suggestions below which I hope the authors will find helpful in improving the manuscript.

We thank the Reviewer for taking the time to evaluate our manuscript and for the constructive suggestions. We respond to each of the comments below.

1. The study found that higher emotion recognition in speech prosody related to better general socio-emotional adjustment. Associations between recognition of vocal emotional expressions and social competence have been examined by previous research in pre-schoolers and school-aged children (see for example studies by Verbeek, 1996, Mitchell, 1995, Baum & Nowicki, 1998, Rothman & Nowicki, 2004, Goonan, 1995, see also Nowicki & Mitchell, 1998 at Genetic Social and General Psychology Monographs etc.). It is not clear what is new information here and what the present study adds to the current knowledge base. It needs to be very clearly explained in the introduction what exactly the present study adds to each previous study that has examined this topic.

We are grateful for the mentioned references. Some of them were missing from our literature review and have now been added to p. 5 (Baum & Nowicki, 1998; Rothman & Nowicki, 2004).

In addition to the large sample size, the main novel aspect of our study is that we cover the two types of nonverbal emotional cues that are present in the human voice: (1) emotional prosody, i.e., changes in 'intonation' as embedded in linguistic content (e.g., speaking with a happy tone of voice); and (2) purely nonverbal vocalizations, such as laughter or crying, often called affective or affect bursts (e.g., Grandjean, 2021, *Emotion Review*).

Research on associations between vocal emotion recognition and social competence to date has focussed on emotional prosody, leaving nonverbal vocalisations unexplored. This is an important limitation because recent studies have shown that we recruit distinct mechanisms for prosody and for nonverbal vocalizations, both in terms of production and perception (e.g., Pell et al., 2015, *Biological Psychology*). And indeed we document a distinct pattern

of associations with socio-emotional functioning for each of them (significant associations for prosody, but not for vocalizations).

The current study is therefore part of the emerging literature that systematically considers nuanced aspects of emotion processing within the auditory modality (prosody vs. vocalizations), in addition to the typical distinction between auditory and visual cues (voices vs. faces). It also contributes to the growing literature on purely nonverbal vocalizations, which remains small compared to the existing body of work on prosody.

The Introduction now provides a stronger emphasis to the distinction between prosody and nonverbal vocalisations (p. 3), and makes a clear case for novelty on p. 6:

'Additionally, inferences have often been based on relatively small samples, typically less than 80 children, and the focus has been on prosody, leaving the other domain of vocal emotional cues – purely nonverbal vocalisations – unexplored. To our knowledge, only one study included nonverbal vocalisations, and the emphasis was on how children matched vocal with facial information (Scheerer et al., 2020). Other poorly understood questions are whether associations between vocal emotion recognition and socio-emotional functioning are specific and direct, or a consequence of general differences in cognitive abilities and socio-economic background. These general factors correlate with both emotion recognition abilities (e.g., Erhart et al., 2019; Izard et al., 2000) and social functioning (e.g., Bellanti & Bierman, 2000; Dearing et al., 2006; Gilman et al., 2003), and are often not considered as potential confounds (e.g., Chronaki et al., 2015; Leppänen & Hietanen, 2001).'

The novel aspects of the study are also highlighted in the Discussion (p. 22-23).

2. In the abstract the following sentence seems to contradict the previous sentence stating that this association was found. 'For emotion recognition in nonverbal vocalisations, no associations with socio-emotional adjustment were found'.

These two sentences do not contradict each other because we have two distinct vocal emotion recognition tasks – one focussed on emotional speech prosody, and a separate one focussed on purely nonverbal vocalisations such as laughter and crying.

We have revised the Abstract for clarity (p. 2):

'The sample included 141 six- to eight-year-old children, and the emotion tasks required them to categorise five emotions (anger, disgust, fear, happiness, sadness, plus neutrality), as conveyed by two types of vocal emotional cues: speech prosody, and nonverbal vocalisations such as laughter.'

[...]

'Based on frequentist and Bayesian analyses, we found that, for speech prosody, higher emotion recognition related to better general socio-emotional adjustment.'

[...]

'For emotion recognition in nonverbal vocalisations, no associations with socio-emotional adjustment were found.'

3. In the introduction (see first page) the authors state 'But it has also been shown that emotion recognition accuracy is higher for nonverbal vocalisations than for emotional prosody'. Please, can the authors explain the distinction between 'nonverbal vocalisations' and 'emotional prosody' as these terms read like they refer to the same construct, nonverbal vocalisations refer to emotional prosody as opposed to verbal vocalisations.

Emotional prosody and purely nonverbal vocalisations indeed correspond to two distinct types of vocal emotional

cues, i.e. they are different constructs. We have revised the Introduction for clarity (p. 3):

'We can communicate vocal emotions via linguistic information, but also via nonverbal cues. Hearing a scream, for instance, can make us realise that someone needs help, or that there is a threat nearby. Nonverbal emotional cues in the human voice can be divided into two domains: inflections in speech, so-called emotional prosody; and purely nonverbal vocalisations, such as laughter and crying, often called affective bursts (e.g., Grandjean, 2021).

Emotional prosody corresponds to suprasegmental and segmental modifications in spoken language during emotion episodes. Prosodic cues include pitch, loudness, tempo, rhythm, and timbre, as embedded in linguistic content (Grandjean et al., 2006; Schirmer & Kotz, 2006). Purely nonverbal vocalisations, on the other hand, do not contain any linguistic information (e.g., screams, laughter), and they represent a more primitive form of communication that has been described as the auditory equivalent of facial expressions (Belin et al., 2004). Prosody and nonverbal vocalisations rely on partly distinct articulatory and perceptual mechanisms (Pell et al., 2015; Scott et al., 2010).'

The Abstract has also been revised accordingly (please see previous comment).

4. I would avoid extensive mention to clinical populations in the introduction if the present study only recruits children from the general population. I understand why the authors cite studies by Nowicki and colleagues and Chronaki et al. as these studies have focused on children from the general population, but it is not clear why the introduction discuss clinical samples (see page 4) without this being the focus of the present study.

We agree, and we have now considerably shortened our mention to clinical populations (from 10 to 6 lines), while also accommodating the request by Reviewer 2 regarding a brief reference to paediatric populations (p. 4-5):

'Studies on clinical populations are suggestive of a link between vocal emotional processing and socio-emotional functioning, both in adult (e.g., Amminger et al., 2012; Jaywant & Pell, 2009; Lima et al., 2013b) and paediatric samples (Deveney et al., 2012; Morningstar et al., 2019; O'Nions et al., 2017). For instance, youth with severe mood dysregulation and bipolar disorder (Deveney et al., 2012), and with depressive symptoms (Morningstar et al., 2019) show impaired recognition of emotional prosody.'

5. Can the authors offer some background information in the introduction about why they expect the association between speech prosody related to better general socio-emotional adjustment to remain significant even when individual differences in age, sex, parental education, and cognitive ability are accounted for. This hypothesis comes as a surprise the reader without any theoretical context in the introduction. The same applies to the 'children completed an additional emotion recognition task focused on facial expressions'. This sentence also comes as a surprise in the last sentences of the introduction as the whole introduction so far has focused on vocal expressions.

We have revised our text as it was indeed too succinct.

On p. 6, we describe previous studies showing that associations between emotion recognition and socio-emotional functioning are sometimes limited to a particular group (e.g., girls), and we also elaborate on why accounting for cognitive and socio-economic factors is relevant:

'Other poorly understood questions are whether associations between vocal emotion recognition and socio-emotional functioning are specific and direct, or a consequence of general differences in cognitive abilities and socio-economic background. These general factors correlate with both emotion recognition abilities (e.g., Erhart et al., 2019; Izard et al., 2000) and social functioning (e.g., Bellanti & Bierman, 2000; Dearing et al., 2006; Gilman et al., 2003), and are often not considered as potential confounds (e.g., Chronaki et al., 2015; Leppänen & Hietanen, 2001).'

Then on p. 7 we introduce our question, and link it directly to the reviewed studies:

'We also examined whether this putative association was limited to a particular group of participants (e.g., girls), or driven by general cognitive and socio-economic factors. In other words, we tested if results remained significant even when individual differences in age, sex, cognitive ability, and parental education are accounted for. This is relevant, considering the reviewed evidence that results can be distinct as a function of sex and age, and that general cognitive and socio-economic factors can be associated with emotion recognition and social functioning, therefore being potential confounds.'

We have not discussed facial expressions in detail throughout the Introduction because they are part of a secondary, exploratory component of our study, and we wanted to keep the text streamlined and short. However, we agree that the mentioned sentence can come as a surprise, and have now expanded the paragraph on p. 7:

More exploratory questions asked which socio-emotional functioning dimensions are more clearly linked to vocal emotion recognition, and whether associations between emotion recognition and social-emotional functioning are specific to the auditory domain, or are similarly seen across sensory modalities. In addition to the two vocal emotion recognition tasks, children also completed an emotion recognition task that focused on facial expressions. There is some evidence that better facial emotion recognition relates to less behavioural problems (Chronaki et al., 2015; Nowicki & Mitchell, 1998; Nowicki et al., 2019) and better self-regulation skills in children (Rhoades et al., 2009; Salisch et al., 2015). But null results have also been reported, namely regarding social avoidance and distress (McClure & Nowicki, 2001), and peer popularity (Leppänen & Hietanen, 2001). Moreover, studies that include the two sensory modalities (i.e., vocal and facial emotions) are relatively rare, and they have also reported mixed findings (e.g., McClure & Nowicki, 2001).

We also mention facial expressions in the opening paragraph of the Introduction on p. 3.

6. In the methods section the authors state that to improve children's engagement throughout the task, an emoji illustrating each emotional category was included both on the response pad and on the laptop screen. Have the authors considered that matching the vocal emotional expression with an emotional facial expression (emoji) may be a confounding variable in recognition?

Thank you for bringing our attention to this aspect. It is challenging to think of a task that is at the same time suitable for children and without potential problems. Providing visual aids might have improved performance and increased the reliance on auditory-visual matching processes; on the other hand, not providing them would make the task less engaging and more reliant on linguistic/reading abilities, which was also not what we wanted.

Our choice was based on previous studies that have used visual aids to optimize emotion recognition tasks for children (e.g., emojis/cartoons in Amorim et al., 2019, and Correia et al., 2019; pictures of people in Sauter et al., 2013). These studies are now mentioned in the Methods (p. 9-10), and task issues are discussed on p. 25:

'Future studies should also extend our findings to different emotion recognition tasks to establish their generalizability. In line with previous studies (e.g., Amorim et al., 2019; Correia et al., 2019; Sauter et al., 2013), we have used visual aids (emojis) to make the task more engaging and less reliant on linguistic/reading abilities, but at the same time this might have inflated performance and increased the reliance on auditory-visual matching processes.'

7. Regarding the methods, could the authors explain the theoretical rationale for which socio-emotional adjustment

(assessed via the CSBQ questionnaire) was completed by the children's teacher only.

The rationale for this methodological choice has been added to the methods as suggested (p. 12):

'Having the teacher completing the questionnaire, instead of a parent, allowed us to maximize sample size, as it could be difficult to get all the 141 parents to return the questionnaire in a timely manner, and to minimize social desirability (for a similar approach, e.g., Leppänen & Hietanen, 2001; Nowicki et al., 2019). Additionally, many of the CSBQ items focus on interactions with peers and behaviours in the school context, which can be best documented by teachers. The teachers were blind to the hypothesis of the study. They had known the children for about one and a half years when they filled the questionnaire, having had the opportunity to interact with them and observe their behaviour on a daily basis.'

8. Please, could you include some brief description in text of the associations between specific emotions and socio-emotional adjustment at the bottom of page 16 rather than only including these results in the table.

This has been done as suggested (p. 18):

'Follow-up multiple regression analyses, conducted separately for each emotion, showed that positive partial correlations could be seen for most emotions, at significant or trend level: happiness, $r = .23$, $p = .01$, $BF_{10} = 3.81$; anger, $r = .22$, $p = .01$, $BF_{10} = 3.20$; fear, $r = .21$, $p = .01$, $BF_{10} = 2.26$; and neutrality, $r = .19$, $p = .03$, $BF_{10} = 1.24$. For sadness and disgust, the trend was in the same direction but did not reach significance: sadness, $r = .12$, $p = .16$, $BF_{10} = 0.30$; disgust, $r = .13$, $p = .12$, $BF_{10} = 0.36$. For completeness, an additional multiple regression was conducted including all emotions simultaneously (see Supplementary Table S6), and none of them contributed uniquely to socio-emotional outcomes ($p_s > .34$), likely because of the shared variance across them.'

9. Page 17 of the results section is not accessible for the reader and needs to be condensed considerably. Please only include a small paragraph with the most important/strongest associations as the key message and include all other statistical details in a table.

We agree, thank you for pointing this out. These results have now been condensed, and all the numbers have been transferred to a new table (Table 2) on p. 18-19.

10. The results of page 17 are difficult to interpret in the absence of a theoretical rationale in the introduction linking specific sub-scales/dimensions of the socio-emotional adjustment questionnaire measure (e.g., prosocial behaviour, behavioural self-regulation, cognitive self-regulation) to vocal emotion recognition. It is not clear why these associations are important and what they mean. Some conceptual framework is necessary in order to understand this.

The analyses of sub-scales are part of a secondary and exploratory component of the study, for which we had no specific hypotheses. Providing a detailed conceptual framework in the Introduction would make the text long and speculative, but we have made revisions for clarity: the Introduction now clarifies that our measure is multidimensional (p. 7), and examples of the covered dimensions are provided, such that the results on p. 18 do not read as a surprise; and we make it explicit that these analyses are exploratory, both on p. 7 and p. 17.

A detailed analysis of the obtained findings, including potential explanations and future directions, is presented in the Discussion (p. 21-23).

11. The discussion needs more work to add some clarity on the issues of theoretical contribution of the present study to the existing knowledge base. Also there are some issues with terminology (e.g. prosody typically refers to nonverbal

vocalisations in the literature as opposed to verbal vocalisations) - these issues also need to be examined more carefully.

We believe that these issues have been addressed by our previous clarifications (please see comments 1-3) - but we would be happy to consider making further changes to the Discussion that the Reviewer might find helpful.

Reviewer 2

This study examines the association between socio-emotional functioning and emotion recognition abilities for prosody, vocalizations, and facial expressions in 6- to 8-year-old children. The authors report that greater socio-emotional adjustment was associated with better recognition of prosody (even after controlling for cognitive ability, age, sex, and parental education), but not of other forms of nonverbal cues. I very much enjoyed reading this paper. The manuscript is very well written and clear. The approach was thorough and sound, and I appreciated the thoughtful discussion of the findings. I have a few questions and suggestions to improve clarity in some parts, and to recommend consideration of alternative interpretations of the findings. Details are below.

We thank the Reviewer for taking the time to evaluate our manuscript and for the insightful suggestions.

1. The authors present evidence about the association between vocal emotion recognition and psychosocial functioning derived from adult clinical populations. However, they omit relevant evidence with pediatric clinical populations (e.g., Manassis & Young, 2000; Emerson, Harrison, & Everhart, 1999; Morningstar et al., 2019; Deveney et al., 2012). Given the age range of interest, it seems more pertinent to refer to this literature than to work with schizophrenia or Parkinson's, for instance.

We agree and we are grateful for the suggested references. Our text on clinical evidence has been shortened (as suggested by Reviewer 1), and it is now more focussed on evidence from paediatric samples (p. 4-5):

'Studies on clinical populations are suggestive of a link between vocal emotional processing and socio-emotional functioning, both in adult (e.g., Amminger et al., 2012; Jaywant & Pell, 2009; Lima et al., 2013b) and pediatric samples (Deveney et al., 2012; Morningstar et al., 2019; O'Nions et al., 2017). For instance, youth with severe mood dysregulation and bipolar disorder (Deveney et al., 2012), and with depressive symptoms (Morningstar et al., 2019) show impaired recognition of emotional prosody.'

2. Hu values should be arcsine-transformed before analyses (Wagner, 1993). Was this step performed?

The arcsine transformation was not justified in our case because analyses of skewness and kurtosis showed no departure from normality, as we report on p. 14: *'there was no substantial departure from normality (skewness, range = -1.38 - 0.75; kurtosis, range = -1.36 - 2.64; Curran et al., 1996.'*

We therefore based the analyses, text and figures on untransformed Hu scores for interpretability. However, we have now repeated the analyses with the arcsine transformation, and could confirm that the results remain unchanged: better socio-emotional adjustment was associated with higher emotion recognition in prosody ($r = .33, p < .001, BF_{10} > 100$), but not in nonverbal vocalisations or facial expressions ($p_s > .23, 0.20 < BF_{10} < 0.23$); and this result holds even when controlling for age, sex, parental education, and cognitive ability (speech prosody: $r = .27, p = .001, BF_{10} = 19.26$; nonverbal vocalisations: $r = .12, p = .18, BF_{10} = 0.27$; facial expressions: $r = .06, p = .52, BF_{10} = 0.13$). The remaining exploratory analyses also pointed to similar results as those reported with untransformed Hu scores.

3. Although averaging across emotions to obtain one Hu score is adequate, many modern studies examine whether there are emotion-specific associations with variables of interest. Given that the evidence the authors review in the introduction and discussion includes emotion-specific links between ER performance and clinical/psychosocial variables, a similar analysis would serve to better contextualize findings within the existing literature. The authors do perform separate models for each emotion type, but this approach ignores the interdependence of ER scores across emotion types. A more thorough investigation of potential emotion-specific links with socio-emotional functioning may be to consider the different emotion types as repeated-measures within a general linear model (which would still allow the authors to include covariates of interest).

We have conducted the suggested model and report the results on p. 47 (Supplementary Table S6). We also mention it briefly on p. 17. None of the emotions uniquely predicted socio-emotional outcomes. Statistically, we believe this is because of the shared variance across emotions. Conceptually, we think this finding corroborates our argument that we are tapping into a general rather than emotion-specific association.

Because we had no predictions regarding specific emotions, we did not elaborate further on this topic. We are happy to conduct further analyses in case we misunderstood the Reviewer's point.

4. More details about the analytical models are needed, particularly regarding the frequentist analyses. What factors were included into the models and how were they operationalized? For instance, when the authors indicate that "performance differed significantly across tasks, $F...$ ", I assume an ANOVA was performed—but I cannot tell whether factors other than 'modality' were entered into the model or controlled for. Including more information about the planned statistical analyses would ensure readers do not have to make assumptions.

We have now added details about our analytic approach to the Data Analysis section (p. 12-13):

'The data were analysed using standard frequentist and Bayesian analyses conducted with JASP Version 0.14.1 (JASP Team, 2020). A repeated-measures analysis of variance (ANOVA) with task (speech prosody, nonverbal vocalisations, and facial expressions) as within-subjects factor was performed to examine differences in emotion recognition across tasks. Pearson correlations and multiple regression analyses were used to test for associations between our variables of interest.'

The specific variables included in the correlations and multiple regression models are specified throughout the Results section for readability (p. 14-19).

5. Minor point: The prosody ER task contained stimuli spoken by female speakers only, whereas the vocalization and facial expression ER tasks included both male and female encoders. Given that speaker gender has been found to influence the recognizability of emotional prosody, could the authors speculate as to whether this aspect of the design could have contributed to differential patterns of results pertaining to the prosody vs. the other two tasks—or note this as a limitation, perhaps?

Good point. We now discuss this possibility as a limitation of our study on p. 25:

'Moreover, the emotional prosody task contained stimuli produced only by female speakers, whereas nonverbal vocalisations and facial expressions included both female and male actors. Because there is some previous evidence that the speaker's sex might influence vocal emotion recognition (e.g., Belin et al., 2008; Zuckerman et al., 1975; but see Amorim et al., 2019), we cannot exclude the possibility this might have contributed to the distinct results across tasks.'

6. I am confused about the application of Holm-Bonferroni corrections or about how it is reported. In the Tables, do asterisks mean that an effect was still considered significant after Holm-Bonferroni was applied? And then, in Table S5, when there are no asterisks... does this mean that none of these p-values were above the corrected Holm-Bonferroni threshold? It is difficult to tell for sure without the p-values being reported.

Yes, in the tables, asterisks mean that the effects remained significant after Holm-Bonferroni correction.

In Table S5 (now Table S6), specifically, we did not in fact correct for the number of multiple regression models conducted (six models in the previous version, now seven because of the new model including all the emotions simultaneously). Because these are follow-up exploratory analyses, conducted primarily to complement the main analyses and to inform future studies, we thought that correcting for the number of models would be too conservative. The same reasoning was applied to the analyses of specific socio-emotional dimensions, also exploratory.

We now make this aspect clear in the Data Analysis section (p. 13) and in the table notes.

7. Hu is lower than I would expect for sadness (in prosody), which was recognized at the same level as disgust in the current study. Typically, sadness is well-recognized in emotional prosody. An inspection of the confusion matrices may be warranted to better understand the source of unexpectedly low accuracy for this emotion.

Confusion matrices are now included (Supplementary Table S2), and they are indeed informative.

Raw hit rates were higher for sadness (49.3%) than for disgust (38.5%) in prosody – but sad and neutral stimuli were often confused with each other (sad stimuli → neutral categorization, 31%; neutral stimulus → sad categorization, 25.9%), and this resulted in the relatively low Hu scores that we report. Confusions sad-neutral are common in the emotional prosody literature (e.g., Castro & Lima, 2010, *Behavior Research Methods*; Chronaki et al., 2015, *Scientific Reports*; Pell et al., 2009, *Journal of Nonverbal Behavior*), and other studies have found that sadness is not always among the best recognized prosodic emotions (e.g., Correia et al., 2019, *NeuroImage*; Pell et al., 2009, *Journal of Nonverbal Behavior*). Here this might have been slightly exacerbated because we tested young children, who are generally less accurate than adults, but the inspection of the confusion matrices suggests that our pattern of results is generally as expected. Thank you for this suggestion.

8. Minor point: some of the text in p. 17 could be put into a Table to make it easier for readers to parse this section and compare the contribution of variables of interest across subscales.

We agree. The text has been substantially shortened and a new table has been included to summarize the corresponding statistics (Table 2).

9. Were children screened for autism spectrum disorders? Autism symptoms could be an important 'third variable' to consider in interpreting findings, given its known association with deficits in vocal ER and socio-emotional functioning.

We did not have a specific measure for autism symptoms, but the parents completed a background questionnaire that asked for information about neurodevelopmental/neurological disorders. This led us to exclude two children, due to a diagnosis of epilepsy, but none of the parents reported autism disorders. Consistent with this, none of the included children had special educational needs/disabilities, as determined by the school system.

We have revised the Methods for clarity, on p. 8:

'All children were Portuguese native speakers and, according to parent reports, had normal hearing and no neurological/neurodevelopmental disorders (e.g., autism spectrum disorders).'

And on p. 12:

'Before the sessions, a parent completed a background questionnaire that asked for information about parental education and employment, and the child's history of health issues, such as psychiatric, neurological/neurodevelopmental disorders, and hearing impairments.'

10. A potential interpretation of the link between behavioural and cognitive SR and prosody ER may be that, compared to vocalisations and still images of faces, emotional prosody requires listeners to 'hold' temporally dynamic information in working memory across longer periods of time to inform interpretation. Self-regulation may thus covary with the type of attention and executive functions that would facilitate the interpretation of prosody, but may not be required as much for other forms of nonverbal cues.

This is a good point that we have now integrated into our discussion of this issue on p. 22:

'In view of evidence that attention can contribute to performance in emotional prosody tasks in adults (e.g., Borod et al., 2000; Lima et al., 2013b) and children (e.g., Filipe et al., 2018), it could have been that children who were more able to focus and remain on task were in a better position for improved performance. For instance, emotional prosody recognition requires listeners to maintain temporally dynamic information in working memory to inform interpretation, and self-regulation may covary with this type of attention (Hoffmann et al., 2012).'

11. Minor point: The authors argue that "discrepancies across studies might stem from differences in samples' characteristics and measures" – such as? Though the authors would need to speculate, extending this discussion could help direct future research.

We have now revised the text along these lines (p. 24):

'These discrepancies across studies might stem from differences in samples' characteristics and measures. For instance, preschoolers (Salisch et al., 2015) compared to school-age children (McClure & Nowicki, 2001), and measures of peer-rated popularity (Leppänen & Hietanen, 2001) compared to measure of social avoidance and distress (McClure and Nowicki, 2001). These discrepancies will be clarified as more research is conducted on this topic.'

We thank again both Reviewers for their insightful comments, which have contributed to what we believe is a significantly improved and stronger manuscript.

Appendix B

iscte

UNIVERSITY
INSTITUTE
OF LISBON

César F. Lima

Assistant Professor

Iscte - University Institute of Lisbon

Avenida das Forças Armadas, 1649-026 Lisboa, Portugal

cesar.lima@iscte-iul.pt

www.cesarflima.com

Lisbon, 12 October 2021

Dr. Teodora Gliga

Associate Editor, *Royal Society Open Science*

Prof. Essi Viding

Subject Editor, *Royal Society Open Science*

Submission RSOS-210454

Associations Between Vocal Emotion Recognition and Socio-Emotional Adjustment in Children

Dear Dr. Gliga and Prof. Viding,

We are pleased to hear that our manuscript has been accepted for publication pending minor revisions.

We resubmit a revised version that addresses the three remaining points raised by the Reviewer. Specifically: (1) we have repeated the main analyses based on arcsine transformed data, and included the findings in Supplementary Materials (p. 16 and p. 50-51); (2) we now make it explicit when inferences are based on uncorrected results (p. 13, 21 and 23); and (3) we mention the null finding for facial emotion recognition in the Abstract. Regarding the last point, we agree that it is a good suggestion, but we had to go slightly over the limit of 200 words to incorporate the new sentence (Abstract word count now = 214). If the limit is strict, we are happy to go ahead with previous version and leave the sentence out.

We look forward to your editorial decision on our article.

Yours sincerely,

César Lima, on behalf of all authors